# Systematic review of oral health in slums and non-slum urban settings of Low and Middle-Income Countries (LMICs): Disease prevalence, determinants, perception, and practices

**Mary E. Osuh**[1,2,3]\*, **Gbemisola A. Oke**[2,3], **Richard J. Lilford**[1,4], **Jackson I. Osuh**[5], **Bronwyn Harris**[1], **Eme Owoaje**[6], **Folake B. Lawal**[2,3], **Akinyinka Omigbodun**[7], **Babatunde Adedokun**[8], **Yen-Fu Chen**[1]

1 Division of Health Sciences, Warwick Medical School, University of Warwick, Coventry, United Kingdom, 2 Department of Periodontology and Community Dentistry, Faculty of Dentistry, College of Medicine, University of Ibadan, Ibadan, Oyo State, Nigeria, 3 Department of Periodontology and Community Dentistry, University College Hospital, Ibadan, Oyo State, Nigeria, 4 Institute of Applied Health Research, College of Medical and Dental Sciences, The University of Birmingham, Birmingham, United Kingdom, 5 Department of Psychology, Faculty of Social Sciences, Federal University, Oye-Ekiti, Nigeria, 6 Department of Community Medicine, Faculty of Public Health, College of Medicine, University of Ibadan, Ibadan, Oyo State, Nigeria, 7 Department of Obstetrics and Gynaecology, Faculty of Clinical Sciences, College of Medicine, University of Ibadan, Ibadan, Oyo State, Nigeria, 8 Centre for Observational Research, Amgen Inc. Thousand Oaks, California, United States of America

\* meosuh@com.ui.edu.ng, Mary.Osuh@warwick.ac.uk, meosuh2@gmail.com

## Abstract

### Background

A comprehensive summary of evidence about oral health in slum settings that could inform policy directions is lacking.

### Objective

To summarise the latest evidence regarding oral disease burden and their determinants, perceptions, practices, and service utilization in the slums and non-slum urban settings of LMICs.

### Design

Systematic review

### Data sources

Embase and MEDLINE (Ovid); PubMed; Scopus, Web of Science, CRD DARE Database; ELDIS; Essential Health Links; HINARI; African Index Medicus (AIM); and Bioline International, all searched from January 2000 to June 2023 using slum-related terms.

**Data Availability Statement:** All relevant data are within the manuscript and its Supporting Information files.

**Funding:** This research was funded by the National Institute for Health and Care Research (NIHR) (16/136/87) using UK aid from the UK Government to support global health research. RJL is also funded from the NIHR Applied Research Collaboration (ARC) West Midlands. YFC is also funded by NIHR Evidence Synthesis Programme, grant number NIHR153453. The views expressed in this publication are those of the author(s) and not necessarily those of the NIHR or the UK Government. The funders had no role in study design, data collection and analysis, decision to publish, or preparation of the manuscript.

**Competing interests:** The authors have declared that no competing interests exist.

### Eligibility criteria

Empirical studies of all designs were eligible. Studies published in English with full-text available and reporting disease burden, perceptions, behaviours and service utilisation related to oral health of residents of slums or broader settings including slums in low and middle-income countries were included.

### Data extraction, quality assessment, synthesis and reporting

Studies were categorised and data were extracted and charted according to a preliminary conceptual framework refined by emerging findings. The Mixed Methods Assessment Tool (MMAT) was used to appraise the quality of empirical studies. The Preferred Reporting Items for Systematic Reviews and Meta-Analyses (PRISMA) guidelines and (where applicable) the Synthesis Without Meta-analysis (SWiM) guideline were adopted for guiding synthesis and reporting. Results were tabulated and narratively summarised.

### Results

Full-text articles for 56 records were assessed for eligibility and 23 of the articles were included in this review. The majority (13 studies, 57%) were conducted in Asia, and nine studies (39%) in Africa. Six focused on slums (two examined slum and urban non-slum and four examined purely slum settings), two examined general urban settings, eight included both rural and urban areas in their settings, two examined disadvantaged/low socioeconomic, one assessed rural/urban/metropolis/municipal/district, three covered the national population or whole country, and one looked at high versus low socioeconomic regions. The commonest oral diseases reported were dental caries (prevalence: 13% - 76%), and periodontal diseases (prevalence: 23% - 99%). These were higher in slum settings and showed differences across age groups, gender, and socioeconomic classes. Most participants in the studies perceived their oral health status as satisfactory, a belief commoner among younger people, males, those in higher socio-economic classes, and employed. Mouth cleaning was mostly once daily, usually in the mornings. The use of toothpaste and brush was commonest. Other oral hygiene implements included toothpowder, chewing-stick, neem, charcoal, sand, snuff, salt, and the fingers. There was widespread engagement in home remedies for oral disease cure or prevention, while the use of professional dental care facilities was generally low and problem-driven.

### Conclusion

The systematic review identified a sparse body of literature on oral health surveys in slums and other urban settings in LMICs. Available data suggest a high oral disease burden, worse in slums, use of inappropriate mouth cleaning tools, self-care practices for pain relief, and few visits to care facilities.

### Systematic review registration

Systematic review registration with PROSPERO in February 2020, number CRD42020123613.

## Introduction

Oral health policymakers, researchers, and care providers require efficiently integrated evidence from large amount of data for rational decision-making [1]. Evidence as to whether research findings can be generalized across populations, settings, and treatment variations, or whether findings vary significantly by particular subsets are key in the decision-making process [1, 2]. Comprehensive summaries of evidence relevant to oral health in slum community settings are scarce. Slum had previously been described as crowded, unhealthy places with a high risk of infection and injury and the residents are often marginalized and have limited access to basic services [3, 4]. The general characteristics of a slum setting include not having access to one or more of the following: durable housing of a permanent nature that protects against extreme climate conditions; sufficient living space which means not more than three people sharing the same room; easy access to safe water in sufficient amounts at an affordable price; access to adequate sanitation in the form of a private or public toilet shared by a reasonable number of people; and security of tenure that prevents forced evictions. These characteristics differentiates the slum from the non-slum urban settings [3, 5]. The slum residents who are socially marginalised and deprived have poorer access to oral health care services, thus increasing the trend of dental diseases among them [4, 6, 7]. Prior to this study, we had conducted a preliminary narrative review (unpublished) and it revealed very few studies conducted on oral health in slum settings globally. In addition, the same exercise revealed a scarcity of literature on community oral health surveys in the slums. This may be due to the technical difficulties and substantial resources required for conducting oral health surveys that would be representative of population groups in slums and other non-slum urban settings, hence the vast majority of published oral health surveys in LMICs have been sporadic and based on convenience samples [8]. However, with the aid of a systematic review, it is possible to gain insight into the oral health issues affecting slums and other urban settings from existing surveys conducted in low and middle-income countries (LMICs).The need to systematically assess literature to summarise the latest evidence regarding oral diseases was therefore established. It is hoped that the outcome would assist to identify knowledge gaps, understand and proffer viable solutions to oral health-related issues, and suggest future research focus in the slums of LMICs.

In this review, we systematically assessed literature on oral disease prevalence, the determinants of oral diseases, the perception, practices and dental service utilization pattern among residents of slums and other urban settings of LMICs to gain insight into the oral health issues affecting them.

## Methods

This systematic review was performed according to current best practice guidance [9, 10]. The broad question of interest was: *What is the prevalence of oral diseases, the determinants, oral health perception, practices and dental service utilization pattern of residents of slum and non-slum urban settings of LMICs?* The protocol for the systematic review was registered with PROSPERO in February 2020 (available from: https://www.crd.york.ac.uk/PROSPERO//display_record.php?ID=CRD42020123613).

## Literature search and study selection

A broad search was made into the following databases (MEDLINE (Ovid); Embase (Ovid); PubMed; Scopus; Web of Science; Centre for Reviews and Dissemination (CRD); Database of Abstracts of Reviews of Effects (DARE); Electronic Development and Environment Information System (ELDIS); Essential Health Links; Health Inter Network Access to Research

Initiative (HINARI); African Index Medicus (AIM); Bioline International) in April 2020 [3] and updated in June 2023. Each search strategy was adjusted to the requirements of each electronic database. Indexed terms and keywords related to the concepts were combined using Boolean operators AND or OR. The databases were searched to identify all relevant articles published from 2000 to June 2023 (with Scopus and Web of Science searched in March 2024). The search strategies [S1 Appendix] followed the structure of [oral health OR dental caries OR (other oral health-related terms)] AND [LMIC related terms] AND [slum and urban-related terms]. We adopted the LMIC search filter developed by Cochrane which compiled their filter based on Word Bank Group classification system for 2021 [11]. In the classification system, countries were divided into four groups based on their gross national income (GNI) per capita: high, upper-middle, lower-middle and low. The upper-middle, lower-middle and low-income countries are classified as LMICs [11]. Other methods utilised for identifying relevant research and grey literature included using the Google Scholar search engine to search the internet, forward and backward citation searches and contacting experts in the field of dental public health in Nigeria. Relevant studies from local and international conferences, e.g., International Association of Dental Research (IADR) and Dental Public Health conferences were also searched.

## The PECO concepts

The key concepts related to the systematic review included oral health, slum and urban setting, and LMICs. The concepts were developed in line with the PECO framework where the Population (P) was represented as the adult residents; the Exposure (E) was the slum setting; the Comparator (C) was the non-slum urban setting while the Outcomes (O) were oral disease prevalence, perceptions and practice related to oral health, oral healthcare-seeking behaviour, and utilization of available dental services.

## Inclusion and exclusion criteria

The papers included fulfilled the following criteria: any study design that provides empirical data on oral disease burden, its determinants, care-seeking behavior, and utilisation of existing services in slums and other urban settings of LMICs. These included quantitative studies that were carried out in a slum or a representative urban population, or national surveys covering such populations. Qualitative studies carried out in slums or other similar urban settings (e.g., low socio-economic areas within cities) were also included, as well as mixed-methods studies with similar coverage. Specifically, studies conducted among adult residents, male and female, who reside in the slums and non-slum urban areas of LMICs were included.

Only full-text articles written in English were included. As our primary focus is on contemporary situation, we restricted the studies to only publications from 2000 to the time of our latest search of main databases study (June 2023). A study was excluded when it was a commentary, opinion or narrative review; focused on rural settings only or other settings that are unlikely to cover areas with slums; and among children-only population groups. Slums in high income countries were excluded because of the different contexts [12, 13]. Studies that were conducted among mentally challenged, disabled, and institutionalised population groups were also excluded.

## Management of records

Records retrieved from the search process were uploaded into Mendeley software for reference management, where duplicates were removed. Retrieved articles were stored on DropBox, a web-based storage service that aided collaboration among the reviewers involved.

## Study selection and data extraction

A total of three reviewers were involved in study selection: two main reviewers (MO and TST) and a third reviewer (YFC) who resolved disagreements. Study selection was carried out independently by the two reviewers, first by inspecting titles and abstracts retrieved from databases, and then by examining full-text papers for those considered potentially relevant, based on the study selection criteria described above. A final decision on the papers to be selected was made through agreements by the two main reviewers following the inspection of the full-texts, and/or by a discussion with the third reviewer, when there was disagreement or uncertainty. Decisions and reasons for excluding articles were clearly recorded by the main reviewers. Calibration exercises were conducted to ensure screening was done consistently to reduce potential errors. At the early stage of screening, the reviewers compared their independent decisions and discussed and resolved inconsistencies, and refined the study selection criteria where needed.

Included studies were mapped to the following sub-review questions:

a. Prevalence of oral diseases

b. Factors associated with oral diseases

c. Attitudes, perception, and belief about oral health status

d. Oral health/ hygiene practices

e. Utilisation of dental services

Data extraction was carried out using a standardised data extraction form (see S2 Table 2 in S1 Table) by the two reviewers independently, in duplicate, and summary tables were additionally checked by the third reviewer. Data extracted from the included studies were as follows: author and year of publication, country of study, the summary of methods deployed, sociodemographic characteristics of the participating population, the exposure, the outcomes, and summary findings. To minimise error during abstraction of data, the reviewers were trained on the process of completing the extraction form by consulting the Cochrane Handbook [14] and receiving feedback from a senior reviewer. Relevant data extracted from individual studies were organised according to sub-review questions mentioned above.

## Quality assessment

Quality assessment was carried out by the two reviewers independently. The Mixed Methods Assessment Tool (MMAT) was used to appraise the quality of the empirical studies included in this research since the review included qualitative, quantitative, and mixed methods studies [15, 16]. Criteria for determining the quality of each study as specified in the standardized MMAT checklist comprised of:

i. quantitative descriptive studies: appropriate sampling strategy, representativeness of the target group, appropriateness of measurements, appreciable low level of risk of non-response bias, and appropriateness of statistical analysis [15].

ii. qualitative studies: the appropriateness of qualitative approach to answer the research question, adequacy of the qualitative data collection methods in addressing the research question, adequate derivation of findings from the data, sufficient substantiation of interpretation of results by data, and coherence between qualitative data sources, collection, analysis, and interpretation [15].

## Strategy for data presentation and synthesis

A narrative synthesis approach was adopted given the predominantly descriptive and quantitative nature of the evidence while maintaining the flexibility to accommodate qualitative evidence [17]. For the review sub-question "attitudes, perception, and belief about oral health status" where a qualitative study was found, we used a convergent/concurrent design as described by Fetters et al.[18] by collecting and analysing data from quantitative and qualitative studies separately in parallel. Integration of the evidence took place during the reporting and interpretation stages. The Preferred Reporting Items for Systematic Reviews and Meta-Analyses (PRISMA) guideline was utilised to layout the study selection process (Fig 1) and to guide reporting [10]. Where applicable, the Synthesis Without Meta-analysis (SWiM) reporting guideline [9] was also adopted. Data from included studies were tabulated and sorted by country, geographical location, setting, and according to issues relevant to this oral health systematic review. These issues were: prevalence of oral diseases, determinants/ risk factors of the diseases, self-perceived oral health status, oral health practices, and self-reported utilization of existing oral health care facilities. Tables were used to facilitate the visualization of data, but no meta-analysis was done because the studies were conducted in different locations and at different times. Given the heterogeneity between these studies, it would be inappropriate to pool all the proportions and rates together. The definition used for a rural / urban settings in this study is in accordance with the United nations recommendations for international reporting, and comparison that characterises settlements based on population size and density [19].

## Exploratory subgroup comparisons

To highlight the unique oral disease burden and healthcare needs of slum residents, particular attention was given to collating data where subgroup comparisons were made between slum settings and non-slum urban settings. Variations in findings between countries and geographical locations and variations in time trends were also noted. To avoid confounding by study-level characteristics, data for these comparisons were primarily obtained from subgroup analyses conducted within individual studies.

## Deviation from the registered review protocol

There were a few deviations from the original protocol registered in PROSPERO. These changes became necessary following observations during the study selection process. Details of the changes are contained in S2 Appendix.

## Results

Using the search strategy described earlier, a total of 13,796 records were identified. Among them, 5,888duplicates were removed, leaving 7,908 abstracts to be screened for relevance. A total of 7,852 records were excluded as they did not meet the inclusion criteria based on titles and abstracts. The remaining 56 full-text articles were then assessed for eligibility. Thirty-three articles were excluded due to the following reasons: commentary (one article); protocol publication (one article); conducted among institutionalized population groups (eight articles); conducted among children/ elderly population exclusively (nine articles); did not include dental health (three articles); based on exclusively rural or village settings (seven articles); high-income country (one article); companion paper–contents were largely overlapping with another included study (three articles). see S2 Table 1 in S1 Table. In total, twenty-three studies were included in the systematic review (Fig 1).

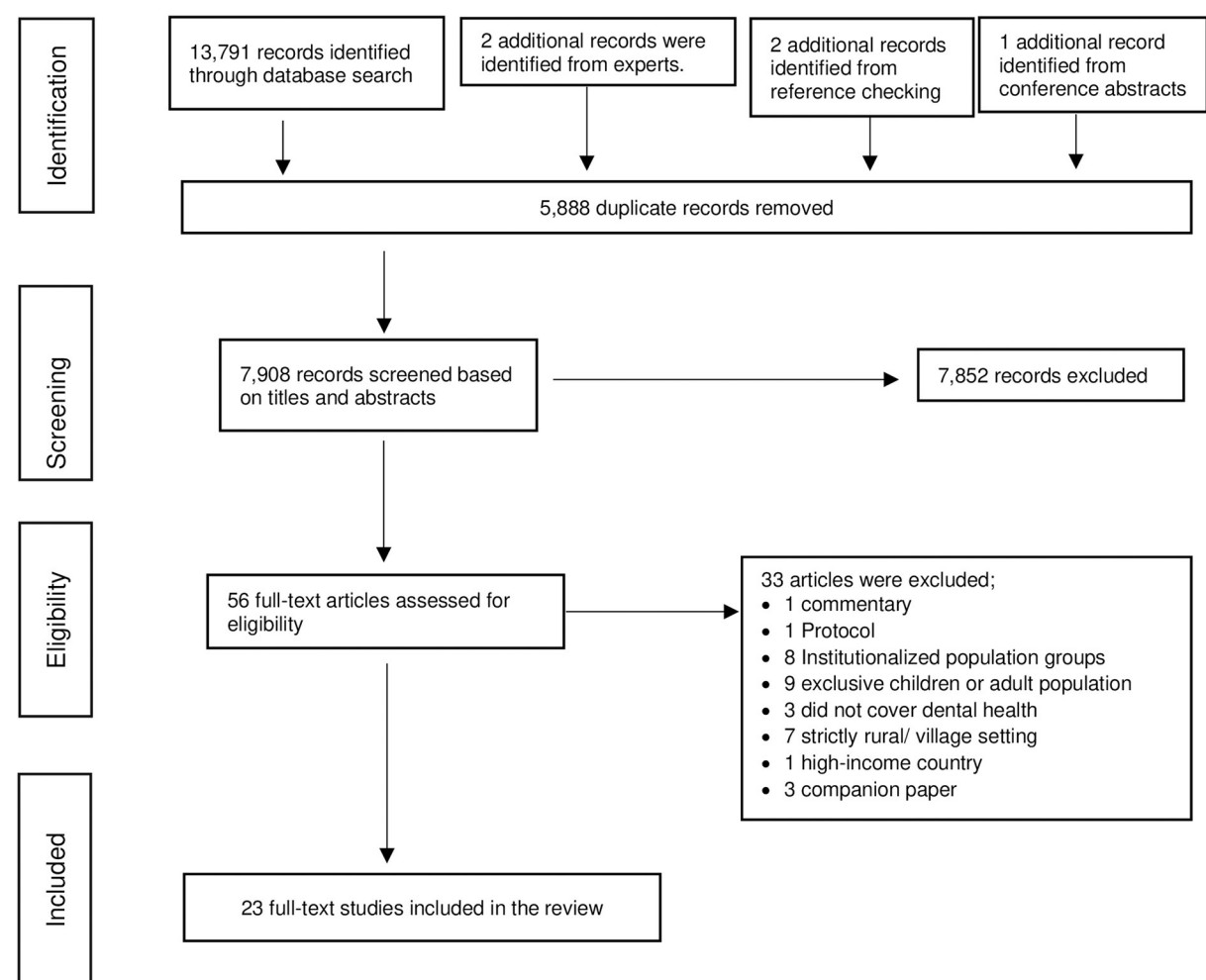

**Fig 1. Summaries of studies retrieved and selected.**

## Summary of characteristics of included studies

All the 23 included studies were primary studies; one of them was a qualitative study and the remaining were quantitative studies. Six of them focused on slums (two compared slums with urban non-slums while four dealt purely with slum settings). Two others examined general urban settings, eight included both rural and urban settings, one assessed rural/ urban/ metropolis/ municipal/ district, three covered entire national population or the whole country, two examined disadvantaged/low socioeconomic areas and one assessed high versus low socio-economic regions. The studies included in the review are summarised in Table 1 according to their settings, methods, participants, and main outcome measures.

The studies were also sorted according to the continents and the countries where the studies occurred. The studies were conducted in fifteen countries in three continents: four from India, one Nepal, two from China, three from Iran, three from Nigeria, and one each from Malawi, Ghana, South Africa, Malaysia, Bangladesh, Rwanda, Tanzania, Burkina-Faso, Pakistan and Brazil. Twelve of the countries are located in Asia, nine in Africa, and one in South America. (S3 Appendix)

**Table 1. Summary of characteristics of included studies in the review.**

| S/N | Setting(s) included / compared | Author & year of publication | Methods | Participants & sample size | Outcome Measures | Code* |
|---|---|---|---|---|---|---|
| 1. | Urban / Urban slum | Patel et al. [20] 2017 | Purposely selected residents of Municipal Corporation (AMC) area of Vejalpur ward, Ahmedabad. Questionnaire and World Health Organization (WHO) oral health surveys and assessment methods (1997) | Indian population / People >10 years of age / N = 300 | Dental caries, staining, abscess formation, mouth ulcers, bad breath, gingivitis, tooth sensitivity, malocclusion, tonsillitis, bleeding gums dental hygiene practices | **a b, d** |
| 2. | Urban / Urban slum | Osuh et al. [4] 2022 | Multistage stratified random sampling of adult residents of a slum and non-slum WHO Oral Health Basic methods WHO oral health survey and assessment methods (2013) | Residents of slum population in Nigeria Adults 18 years and above N Total = 1,357 Slum = 678 Non-slum = 679 | Oral disease prevalence: Caries experience, Periodontal disease Dental erosion Dental trauma Oral mucosal lesions Enamel fluorosis Denture use Level of dental treatment needed (intervention urgency) (Slum vs non-slum sites) Determinants of oral diseases Perception about oral health Oral health practices Utilisation of dental services | **a, b, c, d, e** |
| 3. | Urban slum | Hannan et al. [21] 2014 | A cross-sectional survey was conducted in the 12 slum clusters of Tongi Municipality A close-ended questionnaire was used as well as oral examination using standard indices Oral health assessment method–Not stated | Bangladesh population Participants of all age groups and sex N = 3,904 | Mean DMFT Decayed Missing Filled components of DMFT | **a, b** |
| 4. | Urban slum | Airen et al. [22] 2014 | A cross-sectional survey adopting random sampling technique through the house-to-house visits and enrolment. WHO oral health surveys and assessment methods (1997) | Indore city, Central India Age range 5–64-year-old. 5–14 = (n = 5), 15–34 (n = 86), 35–44 (n = 38), 45–64 (n = 14). N = 143 | Dentition status (caries prevalence and DMFT) and treatment need | **a, b** |
| 5. | Urban slum | Habib et al. [23] 2022 | Cross sectional survey of urban slum clusters in Pakistan through convenience sample WHO guideline using Community Periodontal Index for Treatment Needs (1987) | Rawalpindi, Islamabad region, Pakistan. Adult residents aged between 20 to 50 years N = 385 | presence or absence of gingival bleeding on probing, supra or sub gingival calculus DMFT | **a, d, e** |
| 6. | Urban slum | Chakraborti et al [24] 2023 | Cross sectional survey of urban slum clusters Guideline used to assess oral health status was not provided | Siliguri city, West Bengal, India. Adult slum residents 18 years and above N = 210 | Oral morbidity dental caries, periodontal disease, malocclusion, bleeding, dental pain, oral mucosal lesions, mouth cleaning | **a, b, d** |
| 7. | Urban | Rezaei et al. [25] 2018 | Cross-sectional survey of households to assess dental health-care utilization among household heads in Kermanshah city, western Iran using a self-administered questionnaire | Western Iran population Household head, 18 years and above N = 894 | Utilization of dental services and their determinants | **e** |
| 8. | Urban | Costa et al. [26] 2012 | Home-based, cross-sectional field study WHO Oral health surveys: basic methods (1997) | Brazilian population Adults 35 to 44 years of age N = 1,150 | Caries experience | **a, b** |

(*Continued*)

**Table 1.** (Continued)

| S/N | Setting(s) included / compared | Author & year of publication | Methods | Participants & sample size | Outcome Measures | Code* |
|-----|-------------------------------|------------------------------|---------|----------------------------|------------------|-------|
| 9. | Metropolis/ Municipal/ district/ Urban/ Rural | Hewlett et al. [27] 2022 | Population-based cross-sectional study of adults aged 25 years and above. A random, stratified two-stage sampling method was used to select participants WHO oral health surveys and assessment methods (2013) | Adults aged 25 years and above who were resident in the Greater Accra Region (GAR) of Ghana N = 729 | Prevalence of missing teeth, retained roots, severe periodontitis and poor oral health. Assessing differences within the different localities and districts | **a, b, d, e** |
| 10. | Urban / Rural | Wang et al. [28] 2002 | An epidemiological survey using a Cross-sectional study design on the whole China population using WHO oral health basic methods (1987) | Chinese population Whole population N = 140,712 | Caries experience Periodontal disease: gingival bleeding and calculus | **a, b** |
| 11. | Urban / Rural | Tobin and Ajayi [29] 2017 | Pathfinder survey method stratified cluster sampling technique in 2 Local Government Areas (LGAs) in Kwara State WHO Oral health surveys: basic methods. (1997) | Nigerian population WHO index ages 5–6, 12 and 35–44 years' age groups N = 150 | Dental plaque Calculus Gingivitis Enamel wear Dental caries Mean DMFT. | **a, b** |
| 12. | Urban / Rural | Handa et al. [30] 2016 | Descriptive cross-sectional study using multistage random sampling technique among the population of Gurgaon Block, Gurgaon District, Haryana, India. WHO Oral health surveys: basic methods. (1997) | Indian population WHO's index ages and age groups of 5, 12, 15, 35–44, and 65–74 years N = 810 | Dental health practices Mean DMFT and components Prevalence of dental caries, periodontal diseases malocclusions, dental fluorosis Dental treatment needs | **a, d** |
| 13. | Urban / Rural | Morgan et al. [31] 2018 | WHO Oral Health Surveys Pathfinder stratified cluster methodologies First National Oral Health Survey of Rwanda WHO oral health basic methods (2013) | Rwandan population Whole country 2–5, 6–11, 12–19, 20–39, and 40 and above years N = 2,097 | Prevalence of dental caries. Quality-of-life affectation Access to dental health care | **a b, e** |
| 14. | Urban / Rural | Sun et al. [32] 2018 | Multistage stratified sampling of civilians in all the 31 provinces of China using Questionnaires as well as periodontal health examination was done using Community Periodontal Index (CPI) probe | Chinese population 34–44 years of age N = 4,410 | Periodontal diseases Risk factors or associated factors | **a, b** |
| 15. | Rural / Urban | Masalu et al. [33] 2009 | National pathfinder cross-sectional survey WHO simplified oral health questionnaire for adults | Tanzania, adult respondents from the six geographic zones of mainland N = 1,759 | Oral health-related behaviour and practices | **d, e** |
| 16. | Urban / Rural | Hessari et al. [34] 2007 | Stratified cluster random sampling following the WHO 1997 guidelines Data collected as part of a national survey | All 35- to 44-year-old Iranians living in Iran. N = 8,301 | Dental caries DMFT Bleeding Calculus Shallow pocket Deep pocket | **a, b** |
| 17. | Urban and rural | Varenne et al. [35] 2004 | Multistage cluster sampling of households WHO basic oral health survey 1987 | Final study Burkinabe population covered four age groups: 6 years (n = 424), 12 years (n = 505), 18 years (n = 492) and 35–44 years (n = 493) N = 1,914 | Dental caries prevalence to provide epidemiological data for planning and evaluation of oral health care programmes | **a** |

*(Continued)*

Table 1. (Continued)

| S/N | Setting(s) included / compared | Author & year of publication | Methods | Participants & sample size | Outcome Measures | Code* |
|---|---|---|---|---|---|---|
| 18. | A nationally representative sample of adults | Olutola and Ayo-Yusuf. [36] 2012 | A national representative sample using a multi-stage probability sampling strategy to select 107,987 persons from 28,129 households and obtained living environment characteristics of SASAS participants, including sources of water and energy from the data | South African adults (≥16 years) Participants in the 2007 South African Social Attitude Survey (SASAS). N = 2,907 | Self-rated oral health and associated socio-environmental factors | c, e |
| 19. | Whole country | Msyamboza et al. [37] 2016 | Multi-stage sampling method was used on Enumeration Areas (EAs), households, and participants in the whole country New WHO Surveillance tool 2003. | Malawian population (85% rural) Ages 12, 15, 35–44, and 65–74-year-olds. N = 5,400 | Prevalence of dental caries and missing teeth Risk factors for dental disease. | a b, d |
| 20. | Whole country | Olusile et al. [38] 2014 | Multistage sampling using all 36 states of the federation The questionnaire was developed and refined by the authors with input from dentists in other parts of the country. | Nigeria Ages 18 to 81 N = 7,630 | Perception about oral health status Oral health practices Dental service utilization | c, d, e |
| 21. | Urban disadvantaged / National population | Jaafar et al. [39] 2014 | Random sampling of living quarters (households) in the selected areas of Kuala Lumpur WHO Oral health surveys: basic methods. (1997) | Malaysian population Adults 19 years & older household residents N = 586 | Dental Caries DT, FT, MT, and DMFT Periodontal disease Treatment need Prosthetic (denture wearers) Prosthetic need Overall treatment need | a, b |
| 22. | Disadvantaged group | Singh et al. [40] 2020 | Community based oral health survey involving a stratified random sampling technique. The WHO Basic Oral Health Survey Methods 1997 | Residents of a peri urban area whose people belong to indigenous population of low socioeconomic status comprising ages: 5, 12, 15, 35–44 years & 65–74 years (N = 310) | Caries prevalence Pocket depths Oral hygiene (calculus accumulation) across different age categories. Loss of periodontal attachment across different age groups | a, c, d, e |
| 23. | High and low socioeconomic regions | Gholami et al. [41] 2012 | Purposely selected A qualitative study including focus group discussions; provided an in-depth understanding of individual and group experiences and perception | Residents of Tehran 18 years and above N = 46 | Perception about periodontal illness Attitude to prevention | c, d, e |

*Codes (a, b, c, d, e) refer to corresponding review sub-questions (as listed in 'Study selection and data extraction' Section of the main text) towards which the study has contributed data. a: What is the prevalence of oral diseases (dental caries, periodontal disease, and oral cancer) among adult residents of slum and non-slum urban settings of LMICs? b: What factors are associated with oral diseases in adults residing in the slum and other non-slum urban communities of LMICs? c: What are the perceptions of adult residents of slums and other non-slum urban settings of LMICs towards their oral health status? d: What forms of oral health care practices do adult residents of slums and other non-slum urban settings of LMICs engage in? e: What is the oral health care service utilization pattern of adult residents of slums and other non-slum urban residents in LMICs?; CPI probe: Community Periodontal Index probe; EA: Enumeration Areas; LGA: Local Government Area; SASAS: South African Social Attitude Survey

## Findings from quality assessment of included studies

For each of the 23 included studies, the assessment started with answering three screening questions, followed by questions specific to individual study design.

Screening questions: All included studies were scored "yes" for all being empirical studies, each with clear research question(s) and each having collected data that allowed those research questions to be addressed.

For qualitative studies: Only one study fell into this category. The study set out to explore perceptions of periodontal health and illness and to examine attitudes and beliefs regarding the prevention of gum diseases among Iranian adults [41]. The participants were lay people, 18 years and over and resided in Tehran and were recruited using the purposive sampling. An in-depth understanding of individual and group experiences and perceptions was realised through four focus group discussions which reflected the diversity of socioeconomic levels in Tehran. The assessment of the quality of this study revealed coherence between the qualitative data source, collection, analysis, and interpretation of the data presented. However, the study was weak in external validity because it was female-dominated; consequently, the male's experiences and perceptions might not have been adequately covered. In addition, it was based on purposive sampling method using volunteer participants. Moreover, the study did not give adequate information on the researchers' consideration of their impact on the wider context of the study methods/findings through reflexivity and these may have influenced the study's findings.

Regarding the quantitative descriptive studies, the sampling strategies were relevant to the research questions in all 22 studies. All of the measurements in each of the studies were appropriate for each research question and the statistical analysis adequately answered the research questions. There were variations in the criteria for diagnosis of oral diseases in that only three [4, 27, 31] studies utilised the latest World Health Organization (WHO) guidelines released in the year 2013 for oral disease diagnosis; many other studies utilized earlier versions of the WHO guidelines while a few did not use WHO standardized method. One did not provide information on the oral health assessment guideline used [24]. These variations impacted on the comparability of findings from the different studies to some extent, as there exist subtle differences in the diagnostic criteria and the outcome measures in the different versions of the WHO oral health assessment methods, which formed the basis for revisions. For example, reports on periodontal disease measures for most of the studies were based on earlier versions of the WHO basic oral health survey methods, which measured periodontal status by sextants or index teeth to derive overall prevalence, bleeding, calculus, shallow and deep pockets as outcome measures. Whereas, in the latest WHO version, periodontal status measurement was modified to include assessment of gingival bleeding and recording of pocket scores for all teeth present rather than sextants or index teeth [42]. However, some level of comparability among the reviewed studies was still achieved using overall periodontal disease prevalence.

Regarding the sample representativeness of the target population, the majority of the studies failed to report a response rate; five studies reported response rates which ranged between 84% and 99% [4, 25, 27, 33, 36]. In nine studies a comparison of sociodemographic characteristics such as age and sex between the study sample and the sampling frame from which recruitment was made [24, 26, 30–32, 34, 35, 37, 38, 40], and the results showed no major discrepancies giving a good rating for sample representativeness of the target population and a low-risk rating for nonresponse bias. In three of the studies there was insufficient evidence to show that the sample was representative of the target population, thus making the risk of response bias difficult to assess. Therefore, a decision of "can't tell" was reached. In the first study—conducted in Tanzania [33], two sites from two zones of the six geopolitical zones of the country were purposively selected to represent urban population but the details of the final participant selection process were not clear. In the second study which was conducted in Nigeria [29], the sample size was not calculated, rather a number of subjects were selected in each age group according to the WHO survey method but the recruitment of participants was by volunteering while in the last study which was conducted in Pakistan [23], there was no information on how the samples who were selected through convenient sampling method were recruited. Compliance with the WHO guideline in terms of number of subjects and the non-

random nature of the recruitment process made a clear rating challenging. There was high risk of non-response bias and lack of representativeness of the target population in the sample from one study conducted in Ahmedabad Municipal Corporation (AMC) [20] in which the two wards studied were purposively selected. The risk of non-response bias was uncertain in one study [22]: an unspecified random sampling method was used and the sampling procedure was not clearly described, neither was any information on the response rate or the data required to determine it provided. In the other two studies, the sample sizes were very large and the sampling procedures were clear, which led to a good risk of bias rating [32, 37]. Results of quality assessment for individual studies are presented in S2 Table 3 to S2 Table 7 in the S1 Table.

## Prevalence of oral diseases

Eighteen studies contributed data to this sub-review question. As shown in Tables 1 and 2, only six of the studies specifically involved slum settings [4, 20–23]. The rest were conducted mainly in urban settings with nine of the studies comparing their findings with rural settings [27–35]. Sixteen studies were published after the latest (2013) release in the WHO's series of oral health survey guidelines (manual), twelve of these studies assessed oral disease prevalence, three [4, 27, 31] of which utilised the latest WHO guideline in the examination of oral health (Tables 1 & 2).

The prevalence of dental caries varied across age groups, gender and socioeconomic class and ranged between 13% (Nigeria) [43] and 76% (Central India) [22] of the populations under survey. Overall caries prevalence appeared generally low; however, some highly prevalent cases were observed in India [20, 22, 27], Brazil [26], Malaysia [39] and Rwanda [44], where the prevalence of over 50% were reported. Similarly, within the WHO index age categories, highly prevalent values were reported among 35-44-year-olds in China (63%) [45] and Burkina Faso (73%) [35] and among the 65-74-year-olds in China (65%) [45], and the prevalence appeared to increase with higher index age categories. The Decayed, Missing, Filled Teeth (DMFT), which is a lifetime measure of caries experience of an individual, was usually reported as a mean at population level. The mean DMFT ranged between 0.26–16.6 [21–23, 26, 29–31, 34, 35, 37, 39, 45].

Regarding periodontal disease: In all included studies, periodontal disease affected participants of all ages, gender and socioeconomic standing but was more pronounced among the older age group. Periodontal disease also varied across the gender groups and socioeconomic gradient but was generally high in prevalence, ranging between 23% and 99% and generally higher in prevalence relative to dental caries [4, 21, 23, 24, 27, 30, 32, 34, 39]. Bleeding gum / gingivitis—the early stage in periodontal disease contributed to the highest component in the periodontal disease prevalence value across all studies that examined it as a composite. Of the three studies that compared slum with either non-slum urban settings or national average, the prevalence and severity of most oral diseases were generally higher in the slum settings or urban disadvantaged setting. There is higher prevalence of dental trauma cases in urban settings relative to rural settings and among disadvantaged population groups [43] (Table 2).

Variation in oral disease prevalence within slum/ urban disadvantaged setting versus general urban or national population also featured in Tables 1 and 2. Five studies [4, 20, 22, 39, 40] addressed this review sub-question: Airen et al. 2014 [22] found a higher dental caries prevalence in the slum population (76%) in comparison to that obtained from the national/general population data of 50% - 60%; Both Patel et al. 2017 [20] and Osuh et al. 2022 [4] reported higher oral disease prevalence in the slum sites compared to urban non-slum sites; Singh 2020 [40] found higher disease prevalence among the disadvantaged study population relative to the

**Table 2. Prevalence of oral diseases among adult residents of slum and urban settings of LMICs.**

| S/N | First Author (Year) &Country of study | Population age range / sample size | Setting | Caries — Prevalence | Mean DMFT/ components | Periodontal disease — Overall prevalence | Bleeding gum/ gingivitis | Periodontal attachment loss | Other findings |
|---|---|---|---|---|---|---|---|---|---|
| 1. | Patel et al [20] 2017 India | People >10 years of age, 300 | urban / urban slum | Urban slum = 61% Urban area = 47%. | NA | NA | Bleeding (Urban slum) = 8% Bleeding (Urban area) = 8%. Gingivitis (Urban slum) = 7% Gingivitis (Urban area) = 13%. | N/A | Caries disease was more in slums relative to urban areas |
| 2. | | | | | | | | | Slum vs non-slum |
| 3. | Osuh et al. [4] 2022 Nigeria | Adults 18 years and above N Total = 1,357 Slum = 678 Non-slum = 679 | Urban / urban slum | Slum site = 27% Non-slum site = 23% | Slum site = 1.1 (2.7) Non-slum site = 1.3 (3.1) | Periodontal disease Slum site = 77% Non-slum site = 55% | Gingival bleeding Slum site = 75% Non-slum site = 53% | Attachment loss Slum site = 23% Non-slum site = 16% | Dental erosion 10% vs 7% Dental trauma 32% vs 21% Oral mucosal lesions 3% vs 1% Enamel fluorosis 4% vs 5% Denture use 1% vs 1%. |
| 3. | Hannan et al. [21] 2014 Bangladesh | Participants of all age groups and sex 3,904 | purely slum | NA | Mean DMFT = 3.01 DT = 2.10 MT = 0.91 FT = 0.02 | Prevalence of normal gingiva = 4.1% | Gingivitis = 95.9% | NA | Prevalence of both moderate and severe gingivitis was higher among female than male |
| 4. | Airen et al. [22] 2014 Central India | Age range 5–64 year- 143 | Purely slum Study findings compared to data from general population | 76%, | Mean DMFT was 2.54. MT = 1.29 DT = 1.18 FT = 0.13 | NA | NA | NA | Treatment needs: Restoration -(57.5%) Extraction (32.2%) Pulp care (6.25%) Caries prevalence in study population (76%) was higher than in the general population (50–60%). |
| 5. | Habib et al. [23] 2022 Pakistan | Age range 20–50 years 385 | Urban slum | NA | Mean DMFT = 8.91 ± 7.63 | NA | 21.3% had Bleeding Gum | 33.8% had Pocketing | Mean CPITN = 1.93 ± 0.97. More than a third of study population required emergency levels of dental treatments. |
| 6. | Chakraborti et al [24] 2023 West Bengal, India | Ages 18 years and above 210 | Urban slum clusters | 39.5% | NA | 64.3% | 62.3% | NA | Malocclusion = 26.2% Oral mucosal lesions 16.7% Pain within the past year- 47.4% |

(Continued)

**Table 2.** (Continued)

| S/N | First Author (Year) &Country of study | Population age range / sample size | Setting | Finding / Outcome | | | | | | Other findings |
|---|---|---|---|---|---|---|---|---|---|---|
| | | | | Caries | | Periodontal disease | | | | |
| | | | | Prevalence | Mean DMFT/ components | Overall prevalence | Bleeding gum/ gingivitis | Periodontal attachment loss | | |
| 7. | Singh et al. [40] 2020 Nepal | Peri urban residents belonging to indigenous population of low SES status compared to national data Ages: 5, 12, 15, 35–44years 65–74 years N = 310 | Disadvantaged group | 58% | DMFT age 35–44 = 3.18 ± 5.85 DMFT age 65–74 = 2.40 ± 2.65 | 35 – 44years = 23% 65 – 74years = 44% | 35 – 44years = 3% 65 – 74years = 0% | 35 – 44years = 11% 65–74 years = 25% | | The prevalence of dental caries, periodontal diseases, and prosthetic needs were more when compared to the national data. |
| 8. | Costa et al. [26] 2012 Brazil | Adults 35 to 44 years of age representing adult population 1,150 | Purely urban | Dental caries prevalence = 69% | DMFT ≥ 14 = high severity: mean 16.6, stand. Dev. = 6.973 | NA | NA | N/A | | Caries experience was not different between the eight cities surveyed (p = 0.133). |
| 9. | Hewlett et al. [27] 2022 Ghana | Adults aged 25 years and above who were resident in the Greater Accra Region (GAR) of Ghana N = 729 | Metropolitan/ Municipal/ Ordinary district/ Urban/ Rural | Metropolis = 46% Municipal = 32% District = 41% Urban = 40% Rural = 44% | Metropolis = 2.20 (0.18) Municipal = 1.53 (0.16) District = 1.77 (0.19) Urban = 1.83 (0.12) Rural = 1.81 (0.18) | Metropolis = 46% Municipal = 52% District = 46% Urban = 48% Rural = 44% | Metropolis = 46% Municipal = 86% District = 83% Urban = 87% Rural = 82% | Metropolis = 14% Municipal = 10% District = 16% Urban = 11% Rural = 11% | | There is significant differences in disease prevalence between the different localities and districts |
| 10. | Jaafar et al. [39] 2014 Malaysia | Adults 19 years & older, household residents 586 | Urban disadvantaged Versus National population | Urban 71% national average quoted (90%) | 12.7 DT = 2.66 MT = 8.73 FT = 1.27 | 97.1% | 4.3% Calculus = 47.3% | Pocket 4–5 mm = 22.9% Pocket 6 mm or more = 19.6% | | No evidence to suggest that the overall oral disease burden and treatment need in the urban disadvantaged adult population are higher than the national average, except for periodontal disease |

*(Continued)*

Table 2. (Continued)

| S/N | First Author (Year) &Country of study | Population age range / sample size | Setting | Finding / Outcome | | | | | | Other findings |
|---|---|---|---|---|---|---|---|---|---|---|
| | | | | Caries | | Periodontal disease | | | | |
| | | | | Prevalence | Mean DMFT/ components | Overall prevalence | Bleeding gum/ gingivitis | Periodontal attachment loss | | |
| 11. | Msyamboza et al. [37] 2016 Malawi | 12, 15, 35–44 and 65–74 year old. 5400 | Whole country Urban / rural | Overall caries prevalence = 37.4% 12 year-olds = 19.1% 15 year-olds = 21.9% 35–44 year-olds = 49.0% 65–74 year-olds = 49.2% Overall Missing Teeth = 35.2% Overall Filled Teeth = 6.5% | Mean DMFT = 2.68 among 12 year-olds, 15 year-olds, 35–44 year-olds and 65–74 year-olds = 0.67, 0.71, 3.11 and 6.87 DT = 1.03 MT: 1.54 FT = 0.11 | NA | Bleeding gum prevalence = 23.5% 12 year-olds = 13.0% 15 year-olds = 11.8% 35–44 year-olds = 30.8% 65–74 yr-olds = 36.1% | NA | | Prevalence of dental caries and missing teeth in urban areas were as high as in the rural areas; all p > 0.05. |
| 12. | Morgan et al. [31] 2018 Rwanda | 2–5, 6–11, 12–19, 20–39, and 40 and above years 2,097 | Whole country Combined Urban/ rural sub-group | Overall caries prev. = 54.3%. Rural group exhibited higher caries experience and untreated caries | Mean DMFT = 3.19 DT = 1.36 MT = 1.80 FT = 0.82 | NA | Plaque prevalence = 32.4% Calculus prevalence = 60.0% | NA | | Quality-of-life challenges due to oral diseases/ conditions including pain, difficulty chewing, self-consciousness and difficulty participating in usual activities was reported by 63.9%, 42.2% 36.2%, 35.4% of participants respectively Early dental care was indicated in 61.3% Immediate treatment (urgent relief of pain/ infection) was required by 5.4%. |
| 13. | Wang et al. [28] 2002 China | 5, 12, 15, 18,35–44 and 65–74. 140,712 | National oral health survey Sub-group Urban/ rural | Overall caries prevalence = Not given Among 35-44-year-olds = 63% 65–74-year-olds = 64.8% | Overall Mean DMFT = 4.5 Among 35-44-year-olds = 2.1 65–74-year-olds = 12.4 | NA | Gingival bleeding was 15.7% The mean number of sextants per person with specific periodontal symptoms in adults 35 – 44yrs of age Healthy gingiva = 1.2% Bleeding = 0.4% Calculus = 4.0% | shallow pocket = 18.9% deep pockets = 4.0% Mean number of sextants per person with specific periodontal symptoms in adults 35 – 44yrs of age Shallow pocket = 0.3 Deep pocket = 0.03 | | Gingival bleeding, shallow and deep pockets were lower in the rural sub-group relative to the urban sub-group |

(Continued)

**Table 2.** (Continued)

| S/ N | First Author (Year) &Country of study | Population age range / sample size | Setting | Finding / Outcome | | | | | Other findings |
|---|---|---|---|---|---|---|---|---|---|
| | | | | Caries | | Periodontal disease | | | |
| | | | | Prevalence | Mean DMFT/ components | Overall prevalence | Bleeding gum/ gingivitis | Periodontal attachment loss | |
| 14. | Tobin and Ajayi [29] 2017 Nigeria | 5–6, 12, and 35–44 years' age groups 150 | Urban/ rural | Overall dental caries was 13.0% | Overall Mean DMFT = 0.26 5–6 yrs = 0.30 12 yrs = 0.04 35–44yrs = 0.46 | NA | Plaque = 66% Calculus = 66% Gingivitis = 30% | NA | Overall dental disease prevalence was 91.3%, rural / urban population was 93.3 / 89.3% (p>0.05). There were more cases of trauma (87.5%) seen in urban than rural locations (p = 0.029). |
| 15. | Handa et al. [30] 2016 India | WHO index age groups of 5, 12, 15, 35–44, and 65–74 years 810 | Urban / rural Versus National survey | Overall dental caries prev. = 45% Missing due to caries = 30% Filled teeth = 8% | Overall mean DMFT = 1.61 mean DMFT of 2.49 among urban people in the age group of 35–44 yrs | Periodontal diseases was 65% and is less when compared to the national survey (89%) | Bleeding gum = 15.8% Calculus = 30.7% | Shallow pocket = 5.9% Deep pocket = 12.7% | 46% of the population suffered malocclusions of which 21.2% had the severe type. Dental fluorosis = 46%, of which 11.2% had moderate and 9.6% had a severe type of fluorosis. Treatment was found to be required among 83% of the population |
| 16. | Sun et al. [32] 2018 China | 35–44 years of age 4,410 | Total study population Urban / rural | NA | NA | Overall Periodontal disease = 90.9% | Gingival bleeding = 87.4%, Calculus = 96.7% | Prevalence of pocket 4-5mm = 45.8% deep pocket ≥6mm = 6.9%, clinical attachment loss >3mm = 33.2%. | Prevalence of bleeding, calculus, and pocket depth (PD) = 4- 5mm were slightly higher in the rural group while PD of > 4- 5mm and clinical attachment loss were higher among the urban sub-group |
| 17. | Hessari et al. [34] 2007 Iran | 35–44 years of age 8,301 | Urban/ rural | Decayed Teeth = 24% Filled Teeth = 16% Missing Teeth = 60% | Overall DMFT prevalence ≥1 = 98% Mean DMFT = 11 FT = 1.8 MT = 6.6 DT = 2.6 | Overall Periodontal disease = 99% | Gingival bleeding = 6% Calculus = 40% | Shallow pocket = 43% Deep pocket = 10% | 715 (8%) had 1–3 DMFT, 3,458 (42%) 4–10 DMFT and 3,970 (48%) more than 10 DMFT |
| 18. | Varenne et al. [35] 2004 Burkina Faso | 6 years 12 years 18 years and 35–44 years 1,914 | Ethnic and socioeconomic groups. Households in urban areas and on the recent population census in rural areas. | Age 6years = 38% Age 12 = 29% 18-year = 54% 35–44-years = 73% | Age 12 = 0.7 18-year = 1.9 35–44-years = 6.3 | NA | Score 2 (gingival bleeding and calculus) Age 12 = 57% 18-year = 58% 35–44-years = 49% | NA | prevalence higher in urban than rural areas in all cases No difference in gender for periodontal disease |

AL: attachment loss; DMFT: decayed, missing & filled teeth; DT: decayed teeth; FT: filled teeth; MT: missing teeth; NA: not applicable–outcomes not examined/not reported. PAL: periodontal attachment loss; PD: pocket depth. All prevalent values are combined population settings rates except otherwise stated

national / general population data. Only one study [39] reported no difference in oral diseases prevalence when comparison was made between the disadvantaged study population and the national population.

## Factors associated with oral diseases

Fourteen studies contributed data to this sub-review question (Table 3): Older age, less education, lower income, poor oral hygiene behavior in terms of brushing frequency and regular visits to the dentist as well as disadvantaged residential setting were identified as factors associated with dental diseases. Variation in oral disease prevalence was also associated with residential settings of slum and non-slum urban [4, 20].

The relationship with gender and residential location (urban/ rural, slum / non-slum) as reported in different studies, were multidirectional: while some studies identified a single direction, some identified an opposite direction and others reported that there was not enough evidence to support a relationship. Hewlett et al. [27] reported differences in dental disease prevalence between different settings, noting higher caries prevalence among residents of the ordinary district and higher periodontal disease prevalence among the metropolitan residents. While a study [24] reported a direct relationship between smoking and alcohol use with oral diseases, other studies reported an inverse association with oral diseases [4, 27]. Oral disease was also reported higher among those who failed to clean their mouth after a major meal and those who did not clean their mouth at all [24].

## Attitudes, perception, and belief about oral health status

Six studies [4, 25, 36, 38, 40, 41] contributed data to this sub-review question (Table 4): one qualitative study was conducted among high and low socioeconomic population groups in Tehran [41], five were quantitative studies: one was conducted among a purely urban population [25], one conducted among residents of slum and non-slum [4], one was conducted among disadvantaged population [40] and the last two, across the whole country [36, 38]. From all of these studies, positive perception of oral health status was evident except for study by Rezaei et al 2018 [25] which reported only 39% self-rating their oral health as good. In the qualitative study [41] conducted among high and low socio-economic settings, the participants from both socioeconomic settings were generally satisfied with the value they attached to the maintenance of their oral health and disease prevention. Most participants in the other two studies also had a positive self-rating (good perception) about their oral health status. In one of the studies (South Africa), good perception about oral health status was more pronounced among younger age groups, male gender, higher education and having an employment [36]. See Table 4.

## Oral health/ hygiene practices

A total of eleven studies addressed this review sub-question. Of these, four were conducted in slum settings, one was conducted among disadvantaged population groups, one, a qualitative study, involved high and low socio-economic groups and the rest among general urban population. (Table 5). All included studies reported that most of the different study populations practiced routine hygiene using tooth brush and paste; a few of the studies reported the use of fluoridated toothpaste among participants [4, 37]. Various cleaning implements are deployed for mouth cleaning, notable among which are charcoal and "miswak" in Tanzania [33], cotton wool, chewing stick, salt and water only in Nigeria [38, 46], charcoal, sand, snuff powder, "neem", twang in India and Tehran [30, 37, 41]. Findings from different population subgroups within studies revealed lower use of toothbrush and paste among disadvantaged or

 

**Table 3. Associated factors/ risk factors of oral diseases in adults residing in slum and urban settings of LMICs.**

| S/N | First Author (Year) Country of study | Population age range / sample size | Setting | Finding / Outcome | Slum–urban difference |
|---|---|---|---|---|---|
| 1. | Patel et al. [20] 2017 India | People >10 years of age, 300 | Urban / urban slum | Dental caries was significantly associated with frequency of brushing and habit of mouth gargling in both urban and urban slum (p-value 0.011 and 0.0001): Dental caries being 4.6% and 22.0% among those who brush twice daily and those who don't in the urban area Being 3.3% and 14.0% among those who brush twice daily and those who don't in the slum area. Dental caries being 5.3% and 10.7% among those who gaggle regularly and those who don't in the urban area Being 4.0% and 10.0% among those who brush gaggle regularly and those who don't in the slum area. Incorrect pattern of brushing was also significantly associated with dental caries in both settings (p-value 0.0001 and 0.004). Brushing material and addiction to smoking and tobacco chewing were significantly associated with the development of dental caries. | More in urban slum areas |
| 2. | Osuh et al. [4] 2022 Nigeria | Adults 18 years and above N Total = 1,357 Slum = 678 Non-slum = 679 | Urban / urban slum | Overall, there was a higher prevalence of the key oral health conditions among males except for dental caries. The key oral health conditions were also shown to generally increase with age, and generally reduced with improvement in the level of education and income level both collectively and from the individual study settings. Smoking and alcohol use were found to be inversely associated with oral diseases Prompt and urgent levels of treatment were required for 35% (slum) versus 28% (non-slum) of participants | Odds of having caries were 21% higher for slum dwellers compared to non-slum residents (OR = 1.21, 95% CI: 0.94 to 1.56); and 50% higher for periodontal pocket (OR = 1.50, 95% CI: 1.13 to 1.98), after adjusting for age and sex |
| 3. | Hannan et al. [21] 2014 Bangladesh | Participants of all age groups and sex 3904 | Purely slum | Gender and age are important determinants of dental diseases. Caries was found higher in males. Both decayed and missing components increased and filling components decreased with the progression of age. | NA |
| 4. | Airen et al. [22] 2014 Central India | Age range 5–64 years 143 | Purely slum | Males exhibited a significantly greater number of caries than females (P = 0.008). Severe caries experience was more in those individuals who are not associated with any occupation (P = 0.048) | Caries prevalence was higher in the slums (76%) than in the general population (50–60%) |
| 5. | Chakraborti et al [24] 2023 West Bengal, India | Ages 18 years and above 210 | Urban slum clusters | Oral morbidity was significantly higher among those who didn't clean their mouth, those who didn't clean their mouth after a major meal and those involved in any form of addiction such as alcohol and tobacco | NA |
| 6. | Hewlett et al. [27] 2022 Ghana | Adults aged 25 years and above who were resident in the Greater Accra Region (GAR) of Ghana N = 729 | Metropolitan/ Municipal/ Ordinary district/ Urban/ Rural | Dental diseases increased with increasing age. All dental diseases studied are reported commoner among males except for dental caries which was found higher in prevalence among females Oral diseases were found commoner among participants that were less educated, and receive lower income Smoking and alcohol use were found to be protective on oral diseases | NA |

*(Continued)*

 

**Table 3.** (Continued)

| S/N | First Author (Year) Country of study | Population age range / sample size | Setting | Finding / Outcome | Slum–urban difference |
|-----|-----|-----|-----|-----|-----|
| 7. | Jaafar et al. [39] 2019 Malaysia | Adults 19 years & older household residents 586 | Urban disadvantaged / National population | Age, gender, and race influenced oral diseases. | No difference |
| 8. | Morgan et al.[31] 2018 Rwanda | 2–5, 6–11, 12–19, 20–39, & 40 &> yrs 2,097 | Whole country Combined rural and urban | Caries experience varied significantly with age (increased with increasing age), level of education geographical location (p < .05). Untreated caries varied significantly with age, and geographical location (p < .05) but not by education level (p > .05) | NA |
| 9. | Msyamboza et al. [37] 2016 Malawi | 12, 15, 35–44, and 65–74-year-olds. 5400 | Whole country Rural/ urban | Gender was an important risk factor. Toothache, dental caries and missing teeth were more common in females than males; 46.5 % vs 37.9 %, 40.5 % vs 32.4 %, 37.7 % vs 30.1 % respectively, all p < 0.05. Prevalence of dental caries and missing teeth in urban areas were as high as in the rural areas; 33.3 % vs 37.4 % and 30.9 % vs 33.7 % respectively Location (urban or rural) was not a risk factor for dental caries in this population. | Equal urban-rural settings |
| 10. | Costa et al. [26] 2012 Brazil | Adults 35 to 44 years of age 1,150 | Purely urban | Caries severity was significantly associated with increasing age, visit to the dentist, and lower-income The prevalence of high caries severity among those aged 40 to 44 years was 1.15-fold (95%CI: 1.04 to 1.26) greater than among those aged 35 to 39 years. A greater prevalence of high caries severity was found among those who frequently visited the dentist (prevalence ratio [PR] = 1.18; 95%CI: 1.07 to 1.30) in comparison to those who did not make regular visits to the dentist. A greater prevalence of high caries severity was also found among those with a lower income (PR = 1.11; 95%CI: 1.01 to 1.23) Gender was not associated with dental caries in the multivariate analysis | NA |
| 11. | Sun et al. [32] 2018 China | 34–44 years of age 4,410 | Total study population Urban / rural | Gender (male), educational level, tooth brushing frequency, smoking, dental floss use potentially influences periodontal health status. | NA |
| 12. | Tobin and Ajayi. [29] 2017 Nigeria | 5–6, 12, and 35–44 years' age groups 150 | Urban/ rural | There was no significant difference between having gingivitis, periodontitis, and bleeding gums and the different occupations (p > 0.05). The presence of calculus (p = 0.005) and gingivitis (p = 0.015) was more in males than females. The presence of plaque (p = 0.001) and calculus (p = 0.006) was significantly more among the skilled workforce. There was no significant association between location and presence of an oral condition for all the oral conditions except dental trauma which had a p-value of 0.029, with the urban area having 7 (87.5%) out of the 8 cases. | No rural-urban difference |

*(Continued)*

**Table 3.** (Continued)

| S/N | First Author (Year) Country of study | Population age range / sample size | Setting | Finding / Outcome | Slum–urban difference |
|---|---|---|---|---|---|
| 13. | Hessari et al.[34] 2006 Iran | 35–44 years 8,301 | Urban/ rural | The findings showed significant differences for dental and periodontal indices by socio-demographic factors and educational status Having calculus or pocket was more likely to be present among men (OR = 1.8, 95% CI = 1.6–2.0) and illiterate subjects (OR = 6.3, 95% CI = 5.1–7.8). | |
| 14. | Wang et al[28] 2002 China | 5, 12, 15, 18,35–44 and 65–74. 140,712 | National oral health survey Urban/ rural | Prevalence rates of dental caries in adolescents and young adults tended to be high in urban areas, meanwhile, in old age, the mean caries experience was slightly higher for rural areas. For adults, the caries figures were significantly higher for females than males (P = 0.001) | Variable urban-rural difference |

CI: confidence interval; OR: odds ratio; PR: Prevalence ratio

**Table 4. Perceptions of adult residents of slums and other non-slum settings of LMICs about their oral health state.**

| S/N | First Author (Year) | Country of study | Population age range / sample size | Setting | Finding / Outcome |
|---|---|---|---|---|---|
| 1 | Osuh et al. [4] 2022 | Nigeria | Adult residents 18 and above Slum = 678 Non-slum = 679 1,357 | Slum and non-slum urban setting | Self-reported state of teeth and gums were mostly excellent, very good, good and average. The overall good self-perception of the general oral health state was higher in proportion among the slum residents relative to their non-slum residents' counterpart |
| 2 | Gholami et al. [41] 2012 | Tehran, Iran | Residents of Tehran 18 years and older 46 | High and low socioeconomic regions | Participants considered their oral health as playing an important role in relation to general health and believed the same attention should be paid to oral health problems as to other general health problems They had good attitudes towards the prevention of periodontal disease and oral health |
| 3 | Singh et al. [40] 2020 | Nepal | Ages: 5, 12, 15, 35–44 years & 65–74 years 310 | Disadvantaged population | Perception of the oral health status by the majority of the study participants were reported to mostly range from good to average despite a high prevalence of plaque, calculus, and dental caries among them. |
| 4 | Olusile et al. [38] 2014 | Nigeria | Ages 18 to 81 7,630 | Whole country | Overall, 21% of the participants perceived their oral health status as very good, 37% as good, 27% as fair, 9% as poor or very poor while the remaining were not sure of their oral health status. |
| 5 | Olutola and Ayo-Yusuf. [36] 2012 | South Africa | South African adults (≥16 years) Participants in the 2007 SA Social Attitude Survey (SASAS). 2,907 | A nationally representative sample of adults 16 years and older | 76% (n = 2,067) perceived their oral health status as good Good self-rated oral health was significantly higher among males, younger age group, higher education, and employed respondents Self-rated good oral health was also more common among those who lived in areas that did not have access to basic infrastructure such as piped water or electricity |
| 6 | Rezaei et al. [25] 2018 | West Iran | Household head/ above 18 years 894 | Purely urban | 39% self-rated their oral health as good |

slum populations relative to the rest of the urban population groups that are not disadvantaged or considered non-slum settings [4, 20, 40]. More residents of the urban settings relative to those of the slum /disadvantaged /rural settings, utilised toothbrushes and paste in mouth cleaning. Gholami et al. reported the use of home remedies such as baking soda dissolved in warm water or warm salt rinse to improve gingival health and prevent gum infections [41]. Rural dwellers utilized more of native or indigenous mouth cleaning tools and materials

**Table 5. What forms of oral health care practices do adult residents of slum and other non-slum settings of LMICs engage in?**

| S/N | First Author (Year) | Country of study | Population age range/ sample size | Setting | Finding / Outcome |
|---|---|---|---|---|---|
| 1. | Patel et al. [20] 2017 | India | People >10 years of age, 300 | Urban / urban slum | Brushing materials in the urban area and slum respectively include toothpaste and brush (82% and 68%), toothpowder and brush (5% and 9%), toothpaste and finger (4% and 7%), toothpowder and finger (3% and 5%), others (neem, coal, twang, etc.) (4% and 7%), both indigenous material and toothpaste (1% and 4%). Mouth cleaning is brushing morning only (73% and 83%) or morning and night (27% and 17%). Change of brush was at regular intervals (42% and 29%). Only 25.7% and 11.8% of people used the correct brushing technique. |
| 2. | Osuh et al. [4] 2022 | Nigeria | Adult residents 18 and above Slum = 678 Non-slum = 679 1,357 | Slum and non-slum urban setting | Slum versus Non-slum urban Mouth cleaning of at least twice daily: 24% versus 27% Major mouth cleaning implement used–toothbrush and paste 78% versus 94%. Use of fluoridated toothpaste for brushing: 73% versus 80% Other mouth cleaning implements: chewing stick |
| 3. | Habib et al. [23] | Pakistan | Adult residents 20yrs to 50 years N = 385 | Urban slum | 3.5% did brushing twice (after breakfast and before going to bed) 64.9% brushed just once daily (in morning) 31.6% never brushed rather used other methods like miswak or manjan. Krishnan |
| 4. | Chakraborti et al [24] 2023 | West Bengal, India | Ages 18 years and above residents N = 210 | Urban slum clusters | 28.1% had irregular brushing habits 62.9% did not brush after a major meal |
| 5. | Singsh et al. [40] 2020 | Nepal | Ages: 5, 12, 15, 35–44 years & 65–74 years N = 310 | Disadvantaged population | Mouth cleaning of at least two time daily was practiced by 3% of the population. Toothbrush and paste was the most common teeth cleaning implement among 86% Other teeth cleaning implements include: chewing stick, wooden toothpick, thread and salt. |

*(Continued)*

**Table 5.** (Continued)

| S/N | First Author (Year) | Country of study | Population age range/ sample size | Setting | Finding / Outcome |
|---|---|---|---|---|---|
| 6. | Gholami et al. [41] 2012 | Tehran, Iran | Residents of Tehran 18 years and > 46 | High and low socioeconomic regions | Rubbing the teeth with coal was believed to be a good method of maintaining oral hygiene. Home remedies reported to improve gingival health included using baking soda dissolved in water as well as rinsing with warm salt water for gargling to prevent gum infection or boiled sumac to relieve gum problems. Eating hard fruits and vegetables such as apple and carrot was mentioned as acceptable method for tooth cleaning. *When we forget brushing teeth, for example when travelling, we can eat an apple or carrot before bed, which can clean the teeth sufficiently (a woman aged 35+).* |
| 7. | Hewlett et al. [27] 2022 Ghana | Greater Accra Region (GAR) of Ghana | Adults aged 25 and above 729 | Metropolitan/ Municipal/ Ordinary district/ Urban/ Rural | Mouth cleaning of at least twice daily was practiced by most (69%) of the participants in the following sub-population group distribution (68% vs 71% vs 74% vs 68% vs 77%) Tooth brush and paste was mostly (81%) deployed for teeth cleaning (73% vs 91% vs 85% vs 80% vs 86%) Other tooth cleaning implements: chewing sponge and chewing stick |
| 8. | Msyamboza et al. [37] 2016 | Malawi | 12, 15, 35–44 and 65–74 year olds. 5,400 | Whole country | 39.8% said they cleaned their teeth three times a day, 35.2% said twice, 19.7% said once a day and 2.9% said they never cleaned their teeth. The use of fluoridated toothpaste was reported by 42.6% of the participants. No information was provided on the rural-urban variation in the participants' oral hygiene practices |

(*Continued*)

**Table 5.** (Continued)

| S/N | First Author (Year) | Country of study | Population age range/ sample size | Setting | Finding / Outcome |
|---|---|---|---|---|---|
| 9. | Olusile et al. [38] 2014 | Nigeria | Ages 18 to 81 7,630 | Whole country | Older persons, residents in the northern zones of the country and less educated persons displayed poorer oral hygiene practices. The oral hygiene tool used by the largest proportion of participants was the toothbrush and toothpaste (81% of participants) Other tools used included chewing stick (9.6%), salt (0.6%), water only (0.5%) and cotton wool (0.3%). Some participants (5.6%) reported using multiple tools. Only 10.5% of the participants reported using dental floss or other oral hygiene aids such as mouthwashes. Also, 42.0% of participants reported cleaning their mouths twice daily while 37.1% clean their mouths once a day. |
| 10. | Masalu et al. [33] 2009 | Tanzania | Adult respondents from the six geographic zones of mainland 1759 | Rural / Urban | Nearly 95% of urban and 66.4% of rural residents reported using factory-made toothbrushes with no significant differences between sexes in both settings. Toothpaste was reported to be used by 94.1% of urban and 66.4% of rural residents with no sex difference across localities of residence. The prevalence of charcoal use was 4.6% and 13.2% among urban and rural residents respectively with rural females more likely than males to brush their teeth using charcoal. A higher proportion of rural residents used miswak than their urban counterparts. |
| 11. | Handa et al. [30] 2016 | India | WHO index ages and age groups of 5, 12, 15, 35–44, and 65–74 years 810 | Urban / rural | Results showed that 81.5% (440) of urban and 30.6% (83) of rural respondents in the sample were using toothbrushes and toothpaste, whereas 18% (97) and 49.7% (134) of urban and rural areas respectively used toothpaste or powder with their finger. The use of charcoal, sand, snuff powder, etc., as oral hygiene aids are still moderately prevalent in the rural areas (11.8%) |

relative to their urban counterparts. The frequency of mouth cleaning practice varied among populations within the studies. A smaller proportion of participants in each included study cleaned their teeth at least twice daily (recommended). Similar gradient was observed among slum/ disadvantaged population versus non-slum/ urban populations, rural/ urban population groups in related studies [4, 20, 30, 33]. (Table 5)

### Utilisation of dental services

Ten studies matched the review sub-question (Table 6): one examined the slum and non-slum [4], one assessed the slum population exclusively [23], one assessed a disadvantaged population [40], and one, the rural /urban population [33], while the rest examined the general population. The only qualitative study included examined high and low socioeconomic groups [41]. Where examined, dental service utilization was assessed in terms of "ever utilised" and "utilisation within the preceding 12 months' period". The 'ever utilised' rate ranged between 17% and 71% across all included studies while the utilization rate within a preceding year ranged between 4% and 72%. The pattern of dental service utilisation was generally low, episodic and mostly problem driven. Osuh et al. 2022 [4] found that 13% and 20% of participants in slum and non-slum urban areas respectively felt a need for dental care within a preceding year. Studies by Osuh et al and Singh et al [40] reported a greater proportion of participants that self-reported pain/ discomfort from teeth/mouth in past 12 months [4], yet a much lower proportion accessed care from professional dentists. From the qualitative study, [41] the participants displayed good attitude towards regular dental visits, even though most of their dental visits were for pain relief. Two studies [25, 40] identified a lack of funds as barriers to dental service utilisation. Rezaei et all 2018 [25] also identified income, age, being a university graduate, self-rated poor oral health, and having dental insurance as factors that influenced participants' dental visits (Table 6).

## Discussion

### Statement of purpose and principal findings

Given the lack of contemporary, comprehensive summary of evidence relevant to oral health in slum settings, searches were undertaken of existing surveys conducted in all LMICs, to gain insight into the oral health issues affecting slums and other urban settings. We found few studies on oral health in relation to the slum environment with most from Asia and sub-Saharan Africa. Although the majority did not compare slum and non-slum urban settings, lessons abound. Evidence included in our review suggests that the prevalence and burden of oral diseases vary widely, but are generally high in urban settings in LMICs and worse in slums. The commonest being caries and periodontal disease. Perception of the oral health state was mostly good and the use of inappropriate tooth cleaning materials was rampant. Professional dental care utilisation was generally low and mostly pain driven.

### Findings in the context of existing literature

Despite the small number of eligible studies, the review findings provided some evidence about oral health issues in LMICs from which comparisons can be drawn to support oral health policy decisions on slum and urban settings in LMICs. We discuss each of the key findings below.

### Prevalence of oral diseases

The most common oral diseases reported in the slums of LMICs are dental caries, and periodontal diseases. Dental caries affected all ages but were noted to increase with higher WHO

**Table 6. What is the oral health service utilization pattern of adult residents of slums and other urban residents in LMICs?**

| S/N | First Author (Year) Country of study | Population age range / sample size | Setting | Findings / Outcome | | |
|---|---|---|---|---|---|---|
| | | | | % ever utilized dental services | % Utilized dental service within past year | Others |
| 1 | Gholami et al. [41] 2012 Tehran, Iran | Residents of Tehran 18 years and above 46 | High and low socioeconomic regions | N/A | N/A | The participants considered regular dental check-ups as important; however, further discussions revealed that they usually do not follow this behaviour due to lack of time, laziness, busy lifestyle, cost, and lack of dental insurance. They engaged in alternative care (self-care remedy options) Most of the participants had a positive view toward rinsing the mouth with salt water to cure gingival problems, that is, gum bleeding or looseness *I use a herbaceous drug. I boil sumac and gargle its extract and rub it on my gums. It has a great effect in relieving bleeding and swelling of gums (a women aged 35+).* |
| 2 | Osuh et al. [4] 2022 Nigeria | Adult residents 18 and above Slum = 678 Non-slum = 679 1,357 | Slum versus Non-slum urban setting | 17% versus 24% Overall ever utilization = 20% | Percentage among ever utilized 3% versus 5% Overall = 4% | Pain driven dental visits: 97% vs 99% 13% vs 20% felt a need for dental care within preceding 12 months period 70% vs 67% self-reported pain/discomfort from teeth/mouth in past 12 months Sourced needed care from non-professional dentists: 96% vs 98% |

*(Continued)*

Table 6. (Continued)

| S/N | First Author (Year) Country of study | Population age range / sample size | Setting | Findings / Outcome | | |
|---|---|---|---|---|---|---|
| | | | | % ever utilized dental services | % Utilized dental service within past year | Others |
| 3 | Habib et al. [23] | Adult residents 20–50 years 385 | Urban slum | 14.6% | NA | The 14.6% had visited dentist during emergency only 0.1% visited the dentist every 6 months |
| 4 | Hewlett et al. [27] 2022 Greater Accra Region (GAR) of Ghana | Adults aged 25 and above 729 | Metropolitan vs Municipal vs Ordinary district vs Urban vs Rural | 40% vs 29% vs 22% vs 36% vs 26% Overall ever utilization = 34% | 8% vs 13% vs 11% vs 10% vs 4% Overall utilisation within past year 10% | Pain driven dental visits in 87% distributed in (88% vs 86% vs 92% vs 87% vs 91%) |
| 5 | Singh et al. [40] 2020 Nepal | Ages: 5, 12, 15, 35–44 years & 65–74 years N = 310 | Disadvantaged population | All Males = 23% All Females = 41% | N/A | 61% males vs 63% females self-reported pain/discomfort from teeth/mouth in past 12 months Pain driven dental visits in 70% of all participants 43% male and 57% female deferred recommended dental treatment due to funds. Most (81%) commonly deferred dental treatment is Restorative dental care |
| 6 | Rezaei et al. [25] 2018 West Iran | Household head or 18 years and above 894 | Purely urban | .N/A | 60% and 10% reported visiting a dentist for dental treatment in the past year and for 6-monthly dental check-ups, respectively. | Household income, age, being a university graduate, self-rated poor oral health, and having dental insurance among the study participants influenced visits. 73% self-reported experience of dental pain but did not visit the dentist |

(Continued)

**Table 6.** (Continued)

| S/N | First Author (Year) Country of study | Population age range / sample size | Setting | Findings / Outcome | | |
|---|---|---|---|---|---|---|
| | | | | % ever utilized dental services | % Utilized dental service within past year | Others |
| 7 | Morgan et al.[31] 2018 Rwanda | 2–5, 6–11, 12–19, 20–39, and 40 and above years 2,097 | Whole country Combined rural and urban | 70.6% | NA | Of those who ever visited a dentist, 98.7% sought care because of pain. 64% reported painful aching in the mouth during the preceding year Of those who responded to the question of why they were unable to access care, over half (60.3%) reported that cost was the major reason for not receiving care |
| 8 | Olusile et al. [38] 2014 Nigeria | Ages 18 to 81 7,630 | Whole country | 26% | NA | Older age and more skilled profession associated with ever visiting a dentist 55% of dental visits were for treatment, 24.9% were for check-up, only and the rest for both treatment and check-up. |
| 9 | Masalu et al. [33] 2009 Tanzania | Adult respondents from the 6 geographic zones of mainland 1759 | Rural / Urban | 50% of urban and 37% of rural residents ever visited a dental clinic | NA | More urban females than males more likely to visit the clinic ($p < 0.05$) People with pain had a reduced likelihood of not attending dental treatment and more likely to drink alcohol Common practice of self- medication for oral health problems owing to affordability and accessibility challenges for professional dental services |
| 10 | Olutola and Ayo-Yusuf. [36] 2012 South Africa (SA) | SA adults ($\geq$16 yrs) Participants in the 2007 SA Social Attitude Survey (SASAS) 2,907 | A nationally representative sample of adults 16 years and older | NA | 72% | Those who reported dental attendance in the past year were less likely to have rated their oral health as good than those who did not |

index age categories. The values for the "Missing" component of the DMFT were the highest in all reports, followed by the "Decayed" component. The values for the "Filled" component were the lowest in all measures. These reflected the lack of oral health surveillance and access to dental services, leading to ongoing decay and loss of teeth before treatment could be obtained, highlighting the needs for better access to dental healthcare in LMICs and particularly in slum setting. In future population-based oral health surveys, a second caries-measuring tool–the Significant Caries (SiC) index—might be used together with DMFT to reflect the situation of the most caries-exposed individuals [47]. The information thus provided would enable countries to direct resources to the groups that are worst affected [47, 48].The prevalence of periodontal disease remained high irrespective of a country's income category and there is insufficient evidence to conclude that its prevalence has changed over time [49]. Among the sub-categories of periodontal disease, bleeding gum/ gingivitis formed the highest prevalence in most countries. The fact that these conditions are reversible and self-manageable has implications for targeted oral health education among affected population groups [50] The two studies that compared slum with the non-slum setting [4, 20] reported a higher disease prevalence among the slum dwellers. This pattern of distribution is expected as it mimics the general report of distribution pattern of non-communicable diseases and risk factors such as diabetes mellitus, smoking and obesity, in studies conducted in similar settings [51–53].

The prevalence of dental diseases in the LMICs is comparable to reports from available literature worldwide [49, 54–56]. Other diseases include oral cancers, dental fluorosis (white spots on teeth), dental trauma, and edentulism (tooth loss). While reports from the WHO suggested a decline in dental caries in high-income countries of the world such as the United States and Western and Nordic European countries [57], the disease prevalence has remained steadily high in the LMICs [20, 22, 26, 29–31, 34, 35, 37, 39, 43, 45].

## Factors associated with oral diseases

The associated factors for dental diseases were gender, income level, and location (urban/ rural, slum / non-slum). This finding is supported in the literature as other studies have similarly reported the differential influence of gender on dental diseases [58–64], pitching the female gender at a higher risk of caries and the males at higher periodontal disease risk. Possible explanation for higher dental caries in females includes their earlier eruption of permanent teeth, different salivary composition and flow rate, hormonal fluctuations, dietary habits, genetic variations, and possible links to systemic diseases that interact with caries [58–62]. The observed higher periodontal disease among males [32, 34] is supported by other studies that examined the influence of gender on periodontal disease, thus confirming existing reports on the relationship [63, 64]. The influence of income level and residential location on oral disease may be mediated by the following factors: lack of access to oral health facilities and clean water, and lower purchasing power, which may result in unaffordability of recommended cleaning materials such as the right toothbrush (and reduced frequency of change of toothbrush) and fluoridated toothpaste [65–67].

## Self-perception of oral health state

Most participants in the eligible studies perceived their oral health status to be satisfactory. The finding is consistent with reports of positive perception of general health in related studies [68–71].Good perception about oral health status was more pronounced among the younger age groups, male sex, higher socio-economic class, and having employment. An individuals' perceptions of their oral health can influence their willingness to seek dental care [70] similar to perceptions about health status in general health settings [71, 72]. The fact that good

perceptions about dental health often translate to unwillingness to seek dental care even when these perceptions may not be consistent with the actual state of oral health, raises some concerns. Therefore, it is imperative to focus on policies to promote oral health education to shape the right perceptions about oral health among people in LMICs.

## Oral health care practices

Tooth cleaning is important for oral health and a minimum of twice daily tooth cleaning is the professionally recommended routine to promote individual oral hygiene [73, 74]. In spite of this, tooth cleaning in the slums of LMICs was mostly once and in the mornings, similar to findings from reports of UK [75] and Nigeria [46] population. While thegoal of the tooth cleaning implements deployed in all studies was to keep the mouth free of stain and freshen the breath, some had beneficial effects [76] while others were not hygienic. Access to fluoride is crucial in preventing dental diseases [77, 78], yet there is a sparse of studies on populations' exposure to fluoride from sources like toothpaste, food, water, and applications in the LMICs. This information is necessary for understanding oral health risks in populations and planning public health interventions effectively.

## Utilisation of dental services

Although the need for professional dental care within the preceding 12-month period was indicated among a significant proportion of study participants in various studies, the utilization of dental services was generally low. A similar finding was observed from broader literature on the pattern of utilization of dental services [79–81]. In high-income countries, about 40–80% of the adults would have visited a dentist within one year [82–84]. The poor utilization rate of dental services from the majority of included studies is suggestive of low awareness about oral health among disadvantaged populations in the LMICs. The combination of widespread low utilization of professional dental services and the fact that the few users are mostly driven by pain, is highly suggestive of wide practice of self-care remedy alternatives as well as challenges with affordability and accessibility of dental services [46]. Therefore, the needs to examine the pathway to care among disadvantaged populations in future studies, to bring to bare the structural barriers to the use of dental services to service planners' attention and to explore potential solutions (e.g. publicly funded or private insurance) are strongly indicated. Some of the included studies [4, 41] reported significant practice of home remedies or other self-care options for the treatment of oral diseases as alternatives to visiting a dental professional. Some of these self-care remedy options, which include petrol and vinegar, tobacco, urine, alum, ice-pack, and 'touch and go' herbal remedy, 'over the counter' medicines and battery fluid [41, 85–87] may be potentially hazardous. The reasons attributed to the use of self-care remedies for dental ailments include availability, perceived efficacy, and cost [46, 85, 86, 88]. Moreover, the option of traditional healers for dental disease treatment is freely available at a cheaper cost [46, 85].The belief is, therefore, that a visit is made to the dentist only as a last resort where extreme treatment measures (e.g. extraction) are usually indicated.

## Study limitations and strengths

### Limitations

The term "slum" did not feature in many studies involving oral health. The term "slum" is an evolving phenomenon, which describes specific residential characteristics or settings. It is possible that some previous studies on oral health were conducted in settings similar to slum, yet they were not so labelled [89]. Possibly, authors may have deliberately used other terms such

as informal settlement or deprived urban communities etc. to describe the slum area [90]. Therefore, in the review, we included as many available synonyms as possible for slums both in English and local languages [S1 Appendix] to capture relevant studies, and thus the search yield was greatly expanded. We also searched beyond the main medical databases and the grey literature. We performed reference checking and contacted experts in the field, and utilized studies from conference abstracts.

The variations that exist in the context of slums (classification and feature) both within and between countries in the LMIC regions [89] were not taken into consideration in study selection. Such variations may have influenced the reported oral health outcomes. Furthermore, the heterogeneity in the study objectives and designs made comparisons in the review challenging. Some studies failed to provide details about their population, design, and oral disease indicators. Variations in the measurement of the indicators of oral diseases, in particular, affected comparison between countries. However, deploying MMAT [15] quality assessment tool and data synthesis guided by the SWiM guideline [9], made it easier to compare oral health conditions in slums and urban settings of LMICs. The involvement of two reviewers in making decisions when we were confronted with heterogenous objectives and design of studies minimized bias and enhanced the trustworthiness of study selection process, quality assessment and data extraction. This review was limited to articles written in the English language and restricted to articles published from the year 2000 onwards, hence it is possible that some literature may have been missed. We made some changes to our original protocol as reported in S1 Table. These changes allowed us to be more inclusive of potentially relevant literature given the paucity of studies conducted in slum settings, while being unlikely to introduce particular bias to the review.

## Strengths

To the best of our knowledge, this systematic review is the first in the LMIC to comprehensively explore oral health outcomes in slum settings. The evidence base was expanded by bringing together oral health studies conducted in urban disadvantaged settings relative to the general urban settings in order to inform policy guidelines and direction in reducing oral health inequality. Our study highlighted the paucity of representative population oral health surveys, especially among marginalized or disadvantaged population groups, and contribute to the emerging literature on slum health.

## Implication for research and practice

Epidemiological surveys remain the first step to building efficient systems that can maximise health outcomes [91, 92]. The opportunities for research from this review are broad: As findings indicated paucity of oral health surveys in relation to slum population across LMICs, this review provides a baseline from which further research can advance. Researchers in oral health should be encouraged into studies among people living in disadvantaged settings such as slums relative to other settings using standardised tools such as the latest WHO oral health basic survey manual [42] for oral health assessment and surveillance studies. That way the challenges encountered in the attempt to make comparisons between studies included in this systematic review will be minimised. Using standardized tools may also facilitate future review of studies from different countries and settings, thereby ensuring a meaningful contribution of such studies to a growing evidence base on oral health in slum populations. Oral health stakeholders should work to better position oral health as a priority in public health policies and encourage epidemiological surveys. Oral health stakeholders should consider investing more into health promotion programs to encourage routine dental checks, raise awareness about the

dangers in alternative remedies and self-care for dental pain relief as some of these remedies may pose health hazards and address barriers to accessing oral health services.

## Conclusion

There is paucity of oral health surveys with representative samples, in slums and other urban settings in LMICs, but this review provides a base from which further studies can advance. There is a generally high level of oral diseases in urban settings in LMICs and potentially higher disease burden in slums; evidence of using inappropriate materials/tools for tooth cleaning and symptomatic relief of pain; evidence of high disease burden and low health care utilisation—all suggesting possible needs for both oral health promotion interventions and strengthening of oral health service provision.

## Supporting information

**S1 Appendix. Details of web address and link for each database search strategy details and results for all electronic databases and journals and grey literature.**
(DOCX)

**S2 Appendix. Deviation from the originally registered protocol.**
(DOCX)

**S3 Appendix. Distribution of included studies according to continents, countries and settings.**
(DOCX)

**S1 Table. Supporting tables including list of studies evaluated at full-text screening stage, an example of original data extraction form and completed risk of bias assessments for studies providing evidence for individual sub-review questions.**
(DOCX)

## Acknowledgments

Our sincere appreciation goes out to all the other members of the Improving Health in Slums Collaborative who are not listed as authors but have contributed significantly to the success of this project. The support of Dr Tobi Samuel Tunde-Alao in the retrieval of data is greatly appreciated.

## Author Contributions

**Conceptualization:** Mary E. Osuh, Gbemisola A. Oke, Richard J. Lilford, Jackson I. Osuh, Eme Owoaje, Akinyinka Omigbodun, Yen-Fu Chen.

**Data curation:** Mary E. Osuh, Gbemisola A. Oke, Jackson I. Osuh, Yen-Fu Chen.

**Formal analysis:** Mary E. Osuh, Yen-Fu Chen.

**Methodology:** Mary E. Osuh, Gbemisola A. Oke, Richard J. Lilford, Jackson I. Osuh, Folake B. Lawal, Babatunde Adedokun, Yen-Fu Chen.

**Project administration:** Mary E. Osuh, Jackson I. Osuh.

**Resources:** Mary E. Osuh, Richard J. Lilford, Jackson I. Osuh, Akinyinka Omigbodun, Yen-Fu Chen.

**Software:** Mary E. Osuh, Yen-Fu Chen.

**Supervision:** Gbemisola A. Oke, Richard J. Lilford, Bronwyn Harris, Eme Owoaje, Yen-Fu Chen.

**Validation:** Mary E. Osuh, Gbemisola A. Oke, Richard J. Lilford, Jackson I. Osuh, Bronwyn Harris, Eme Owoaje, Folake B. Lawal, Akinyinka Omigbodun, Babatunde Adedokun, Yen-Fu Chen.

**Visualization:** Mary E. Osuh, Gbemisola A. Oke, Richard J. Lilford, Jackson I. Osuh, Bronwyn Harris, Eme Owoaje, Folake B. Lawal, Akinyinka Omigbodun, Babatunde Adedokun, Yen-Fu Chen.

**Writing – original draft:** Mary E. Osuh, Gbemisola A. Oke, Jackson I. Osuh, Bronwyn Harris, Eme Owoaje, Folake B. Lawal, Akinyinka Omigbodun, Babatunde Adedokun, Yen-Fu Chen.

**Writing – review & editing:** Mary E. Osuh, Gbemisola A. Oke, Richard J. Lilford, Jackson I. Osuh, Bronwyn Harris, Eme Owoaje, Folake B. Lawal, Akinyinka Omigbodun, Babatunde Adedokun, Yen-Fu Chen.

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
