## [Decision Letter · Decision Letter 0]

22 Jan 2024

PONE-D-23-35856Systematic review of oral health in slums and non-slum urban settings of Low and Middle-Income Countries (LMICs): disease prevalence, determinants, perception, and practicesPLOS ONE

Dear Dr. Osuh,

Thank you for submitting your manuscript to PLOS ONE. After careful consideration, we feel that it has merit but does not fully meet PLOS ONE’s publication criteria as it currently stands. Therefore, we invite you to submit a revised version of the manuscript that addresses the points raised during the review process.

We look forward to receiving your revised manuscript.

Kind regards,

Easter Joury

Academic Editor

PLOS ONE

Journal Requirements:

This research was funded by the National Institute for Health and Care Research (NIHR) (16/136/87) using UK aid from the UK Government to support global health research. RJL is also funded from the NIHR Applied Research Collaboration (ARC) West Midlands. YFC is also funded by NIHR Evidence Synthesis Programme, grant number NIHR153453. The views expressed in this publication are those of the author(s) and not necessarily those of the NIHR or the UK Government.

Reviewers' comments:

Reviewer's Responses to Questions

**Comments to the Author**

1. Is the manuscript technically sound, and do the data support the conclusions?

Reviewer #1: Yes

Reviewer #2: Yes

2. Has the statistical analysis been performed appropriately and rigorously? 

Reviewer #1: No

Reviewer #2: N/A

3. Have the authors made all data underlying the findings in their manuscript fully available?

Reviewer #1: Yes

Reviewer #2: Yes

4. Is the manuscript presented in an intelligible fashion and written in standard English?

Reviewer #1: Yes

Reviewer #2: Yes

5. Review Comments to the Author

Reviewer #1: This systematic review evaluated the disease prevalence, determinants, perception, and practices of oral health in slum and non-slums urban settings of LMICs. My comments and concerns are below:

• Authors should provide the specific definition for key terms used throughout the study. These include definitions for:

o Slum and non-slum settings

o Urban and rural settings

• Narrative review work mentioned in the introduction should be referenced. If not published, authors should declare that the work is “unpublished.”

• What classification criteria (e.g., World Bank etc.) was used to identify the countries that formed LMICs in the study? How many countries in total? What year was the classification criteria introduced/used by the authors?

• Abbreviations for less common electronic databases like CRD DARE and ELDIS should be accompanied with full meanings.

• The PECO framework referred to that guided the inclusion of the study should be clearly stated as sub-headings for clarity.

• Was a mixed-methods synthesis approach used for study synthesis in this review? Since it includes both quantitative and qualitative studies, how was synthesis done? I see that only a study is qualitative in nature. Authors may consider excluding this study to reduce heterogeneity of study methods in the included study. Else, authors should consider the mixed methods synthesis approach which may be difficult given one qualitative study.

Reviewer #2: Thanks to the authors for the effort spent researching and writing this article. However, there are many questions about the study, as follows:

- Why the systematic review was chosen as a study design? Are there other designs that are more appropriate (e.g. scoping review)? Please justify.

- Please add to the introduction some definitional sentences about the slum environment in a manner consistent with the definition mentioned in the protocol published in PROSPERO.

- As for databases and literature searches, what about Scopus, Web of Science, and other databases? How is it ensured that no articles could be included in the current review in these databases?

- You mentioned that you used calibration exercises to refine the screening process. Could you please explain in detail?

- The researchers mentioned in the Materials section: "Only full-text articles written in English and publications from 2000 to June 2023 were included.". Wouldn’t it have been better to expand the scope of the search to include articles published before 2000, since the subject studied is not a recent topic, and therefore there may be articles about it within the nineties?

6. PLOS authors have the option to publish the peer review history of their article (what does this mean?). If published, this will include your full peer review and any attached files.

Reviewer #1: No

Reviewer #2: No

---

## [Author Response · Author response to Decision Letter 0]

3 Apr 2024

Reviewer 1: 

Review Comment-: “This systematic review evaluated the disease prevalence, determinants, perception, and practices of oral health in slum and non-slums urban settings of LMICs. My comments and concerns are below:

• Authors should provide the specific definition for key terms used throughout the study. These include definitions for:

 o Slum and non-slum settings

 o Urban and rural settings”

Response: The authors note this comment with appreciation.

We have now included the definition of slum and non-slum urban settings as well as Urban and rural settings in the body of the text. These can be found in the following locations: 

Slum/ non slum urban settings: Page 5 lines 104 – 112 under the subtitle Introduction

Rural/ Urban settings: Page 11 lines 261 – 263 under the subtitle- Strategy for data presentation and synthesis

• Narrative review work mentioned in the introduction should be referenced. If not published, authors should declare that the work is “unpublished.”

Response: The authors note this comment with appreciation

The narrative review work mentioned is not published and so has been designated so. This appears on Page 5 lines 114 – 116.

The statement now reads: 

Prior to this study, we had conducted a preliminary narrative review (unpublished) and it revealed very few studies conducted on oral health in slum settings globally…

• What classification criteria (e.g., World Bank etc.) was used to identify the countries that formed LMICs in the study? How many countries in total? What year was the classification criteria introduced/used by the authors?

Response: The authors note this comment with appreciation

The classification criteria used to identify countries that formed LMICs in this study and the year the classification criteria were introduced have now been included in the text. It can be found on Page 7; lines 157 – 162.

The statement reads: For the definition for the Low and Middle Income countries (LMICS) we adopted the LMIC search filter developed by Cochrane which compiled their filter based on Word Bank Group classification system for 2021 [11]. In the classification system, countries were divided into four groups based on their gross national income (GNI) per capita: high, upper-middle, lower-middle and low. Therefore, the upper-middle, lower-middle and low-income countries are classified as LMICs [11].

• Abbreviations for less common electronic databases like CRD DARE and ELDIS should be accompanied with full meanings.

Response: The authors note this comment with appreciation. 

We have now accompanied the acronyms with their full names within the text and can be found in Page 7, lines 145 – 150. 

It reads: …. the following databases (MEDLINE (Ovid); Embase (Ovid); PubMed; Centre for Reviews and Dissemination (CRD) Database of Abstracts of Reviews of Effects (DARE); Electronic Development and Environment Information System (ELDIS); Essential Health Links; Health InterNetwork Access to Research Initiative (HINARI); African Index Medicus (AIM); Bioline International) in April 2020 [3] and updated in June 2023.

• The PECO framework referred to that guided the inclusion of the study should be clearly stated as sub-headings for clarity.

Response: The authors note this comment with appreciation

We have now stated clearly the PECO framework that guided the inclusion and included as a subheading for better clarity. These changes can be found in Page 8 Lines 170 – 176. 

• Was a mixed-methods synthesis approach used for study synthesis in this review? Since it includes both quantitative and qualitative studies, how was synthesis done? 

Response: The authors note this comment with appreciation 

We expected that qualitative studies would be found mainly for the review sub-question of “attitudes, perception, and belief about oral health status”, and therefore adopted a narrative synthesis approach for the systematic review for synthesising the predominantly descriptive and quantitative evidence with the flexibility to accommodate qualitative evidence. As only one qualitative study was eventually identified and included in the review, we used a convergent/concurrent design as described by Fetters et al. by collecting and analysing data from quantitative and qualitative studies separately in parallel, and for integration of the evidence to take place during the reporting and interpretation stage of the review. 

We have now revised the text to reflect the method used in the synthesis and the reference is also provided. 

The statement now reads:

“A narrative synthesis approach was adopted given the predominantly descriptive and quantitative nature of the evidence while maintaining the flexibility to accommodate qualitative evidence [16]. For the review sub-question “attitudes, perception, and belief about oral health status” where a qualitative study was found, we used a convergent/ concurrent design as described by Fetters et al. [17] by collecting and analysing data from quantitative and qualitative studies separately in parallel. Integration of the evidence took place during the reporting and interpretation stages”. Available on Page 11, lines 245 - 251

• I see that only a study is qualitative in nature. Authors may consider excluding this study to reduce heterogeneity of study methods in the included study. Else, authors should consider the mixed methods synthesis approach which may be difficult given one qualitative study.

Response: We thank the reviewer for the suggestions. While we agree that the inclusion of a qualitative study increases the methodological heterogeneity of evidence included in our review, qualitative study is an appropriate study design to address our review sub-question of ‘Attitudes, perception, and belief about oral health status’. As our review team included researchers with qualitative research expertise, the slight increase in methodological heterogeneity in the included studies and corresponding complexity in synthesis method did not cause any particular problems, and we believe that data from the qualitative study complemented data from quantitative studies well and have enriched the synthesis as a whole. Therefore, we do not think that it is necessary to exclude the qualitative study. We are aware of more sophisticated methods for the integration of qualitative and quantitative evidence; nevertheless, as there was only a single qualitative study, we felt that our approach to synthesising quantitative and qualitative findings through juxtaposing the data and then making comparisons and contrast at reporting and interpretation stages is sufficient for the purpose of this review while avoiding over-complicating the synthesis methods.

Reviewer 2: 

Review Comment-: Thanks to the authors for the effort spent researching and writing this article. However, there are many questions about the study, as follows:

- Why the systematic review was chosen as a study design? Are there other designs that are more appropriate (e.g. scoping review)? Please justify.

Response: We have chosen the systematic review design for a couple of reasons: (1) our initial scoping of the literature indicated that the number of related studies is likely to be small; (2) we not only intended to know the volume and nature of relevant literature, but also wanted to appraise and summarise currently available evidence. A ‘standard’ systematic review is therefore more suitable than a scoping review in this context. We were aware that our research questions were relatively broad, and therefore adopted a flexible approach to assessing and synthesising evidence, e.g. by using the narrative synthesis approach and the MMAT tool which accommodates studies of different designs. 

- Please add to the introduction some definitional sentences about the slum environment in a manner consistent with the definition mentioned in the protocol published in PROSPERO.

Response: The authors note this comment with appreciation. We have now included into the introduction, additional definitional sentences consistent with the published protocol, about the slum environment. 

It reads: The slum residents who are socially marginalised and deprived have poorer access to oral health care services, thus increasing the trend of dental diseases among them [4, 6, 7]. Page 5 Lines 112 - 114

The earlier definitional statement which had addressed the 1st reviewer comment is in Page 5; lines 104 - 112

- As for databases and literature searches, what about Scopus, Web of Science, and other databases? How is it ensured that no articles could be included in the current review in these databases?

Response: We have conducted extensive searches using health-related databases where studies relevant to this review are most likely to be found.

In response to the peer reviewer’s comment, we have now extended our database search to include Scopus (with a limit to the subject area of dentistry) and Web of Science. Retrieved records were mostly duplicates. We found a single additional study that met our inclusion criteria and have now incorporated this study into our review. The corresponding changes have also been effected in all affected sections of our manuscript from abstract through conclusions. 

- You mentioned that you used calibration exercises to refine the screening process. Could you please explain in detail?

Response: The authors note this comment with appreciation.

We have now expanded this statement to provide details on how the calibration exercise was done to refine the screening process. This can be found in Page 9; line 210 to 213. 

The statement now reads: Calibration exercises were conducted to ensure screening was done consistently to reduce potential errors. At the early stage of screening the reviewers compared their independent decisions and discussed and resolved inconsistencies, and refined the study selection criteria where needed.

- The researchers mentioned in the Materials section: "Only full-text articles written in English and publications from 2000 to June 2023 were included.". Wouldn’t it have been better to expand the scope of the search to include articles published before 2000, since the subject studied is not a recent topic, and therefore there may be articles about it within the nineties? 

Response: The authors note this comment with appreciation. 

Although more studies might be located by extending the literature search before year 2000, findings from these studies would be less relevant to contemporary situation, which is our primary focus. We therefore believe that the year limitation is justifiable considering the additional time and resources that would be required. We have acknowledged this limitation. 

The statement can be found in page 38; line 652-654 which reads: This review was limited to articles written in the English language and restricted to articles published from the year 2000, hence it is possible that some literature may have been missed. 

We have also justified the reason for our time restriction in the inclusion and exclusion criteria. It reads: Only full-text articles written in English were included. As our primary focus is on contemporary situation, we restricted the studies to only publications from 2000 …onwards 

Page 8; lines 187 - 189

---

## [Decision Letter · Decision Letter 1]

2 Jul 2024

PONE-D-23-35856R1Systematic review of oral health in slums and non-slum urban settings of Low and Middle-Income Countries (LMICs): disease prevalence, determinants, perception, and practicesPLOS ONE

Dear Dr. Osuh,

Thank you for submitting your manuscript to PLOS ONE. After careful consideration, we feel that it has merit but does not fully meet PLOS ONE’s publication criteria as it currently stands. Therefore, we invite you to submit a revised version of the manuscript that addresses the points raised during the review process.

**ACADEMIC EDITOR: **

**I commend on your efforts in revising the manuscript, however there are still some minor queries raised by the reviewers'. Kindly respond to those to make your manuscript more robust.**

We look forward to receiving your revised manuscript.

Kind regards,

Tanay Chaubal

Academic Editor

PLOS ONE

Journal Requirements:

Reviewers' comments:

Reviewer's Responses to Questions

**Comments to the Author**

1. If the authors have adequately addressed your comments raised in a previous round of review and you feel that this manuscript is now acceptable for publication, you may indicate that here to bypass the “Comments to the Author” section, enter your conflict of interest statement in the “Confidential to Editor” section, and submit your "Accept" recommendation.

Reviewer #2: All comments have been addressed

Reviewer #3: All comments have been addressed

2. Is the manuscript technically sound, and do the data support the conclusions?

Reviewer #2: Yes

Reviewer #3: Yes

3. Has the statistical analysis been performed appropriately and rigorously? 

Reviewer #2: N/A

Reviewer #3: Yes

4. Have the authors made all data underlying the findings in their manuscript fully available?

Reviewer #2: Yes

Reviewer #3: Yes

5. Is the manuscript presented in an intelligible fashion and written in standard English?

Reviewer #2: Yes

Reviewer #3: Yes

6. Review Comments to the Author

Reviewer #2: The authors have satisfactorily addressed my comments raised in the previous round of review, and I believe that this manuscript is now acceptable for publication.

Reviewer #3: The strenuous efforts of the authors to conduct this comprehensive review (Suggestion: Lines 611-612-we should avoid absolute terms like these as authors; the review looks like a bird-eye view of oral health situation in urban slums) are appreciable.

However, I would recommend to the authors to address three main issues:

Reducing the article length as it may be a cause of boredom for the readers.

In the methodology section, please bring together three screening questions (Lines 285-290), PRISMA, SWiM (for study selection Lines 213-214), MMAT (Line 199 for assessing the studies’ quality) in that order to make the things more sequential.

Discussion from Lines 512-582 takes the form of results which should be avoided. I would suggest that it should mainly take up:

The salient review’s findings and their implications for the policy makers/ planners and researchers.

The limitations.

The impact of the changes in the registered protocol, if any, should be mentioned.

Age group distribution; and decayed, missing and filled components of dental caries-implications???.

Gingivitis/ bleeding gums (reversible and self-manageable conditions). The need for oral health education should be recommended).

The use of fluoride, explored in very few studies, should be stressed upon.

Structural barriers to the availability, accessibility and use of dental services (publicly funded/ private, insurance etc.) should be brought to planners' attention.

The last three are related to lines 114-116 which need to be rephrased as well.

In addition, in Line 192, exposure/ outcome (the terms more related to the experimental studies) are confusing???

Lines 193-194/ 230-232: Who trained the reviewers and how???

Words to replace: Lines 246-accessed or assessed? Lines 402-403 regular dental visits or irregular ones???

7. PLOS authors have the option to publish the peer review history of their article (what does this mean?). If published, this will include your full peer review and any attached files.

Reviewer #2: **Yes: **Eiad Zinah

Reviewer #3: **Yes: **ABDUL HALEEM

---

## [Author Response · Author response to Decision Letter 1]

23 Jul 2024

Reviewer #2: 

Review Comment-: The authors have satisfactorily addressed my comments raised in the previous round of review, and I believe that this manuscript is now acceptable for publication.

Response: The authors note this comment with appreciation.

Reviewer #3:

Review comment:  The strenuous efforts of the authors to conduct this comprehensive review (Suggestion: Lines 611-612-we should avoid absolute terms like these as authors; the review looks like a bird-eye view of oral health situation in urban slums) are appreciable.

Response: We thank the reviewer’s recognition of our efforts in the preparation of this review. The line numbers quoted in Reviewer 3’s comments above and below do not seem to correspond to the line numbers appearing in the version of the manuscript that we submitted and the PDF file created by the manuscript submission system. We have tried our best to match the reviewer’s comments with our manuscript content, but the discrepancies in line numbers made it quite challenging for us to respond to some of the comments. We would appreciate further clarifications from the reviewer or the editor in case any comments are considered to be insufficiently addressed. 

Review comment:  However, I would recommend to the authors to address three main issues: Reducing the article length as it may be a cause of boredom for the readers.

Response: We have attempted to shorten the manuscript as the reviewer recommended. We are hesitant to remove some details as they were required to ensure completeness and transparency of reporting, including information requested by other peer reviewers. We are happy to move some content (e.g. details of quality assessment of included studies) to Supplementary Appendices if deemed necessary by the editors. 

Review comment:  In the methodology section, please bring together three screening questions (Lines 285-290), PRISMA, SWiM (for study selection Lines 213-214), MMAT (Line 199 for assessing the studies’ quality) in that order to make the things more sequential.

Response: We are not sure what ‘three screening questions’ the reviewer was referring to. We believe that PRISMA, SWiM and MMAT are described in the most suitable places in the manuscript. These may be found under the following two sections: - Quality assessment (Page 10; Lines 226 - 239); and 

-Strategy for data presentation and synthesis (Page 11; Lines 241 – 260)

Review comment:  Discussion from Lines 512-582 takes the form of results which should be avoided.

Response: Thank you. We have revised the text to minimise repetition of findings already presented in the Results section. We have also moved some texts to the Results section where they are more suitably located. 

 For example: the statement “Various cleaning implements are deployed for mouth cleaning, notable among which are charcoal and “miswak” in Tanzania [33], cotton wool, chewing stick, salt and water only in Nigeria [38, 46], charcoal, sand, snuff powder, “neem”, twang in India and Tehran [30, 37, 41]” was moved from the discussion section to the Results section (Page 29; Lines 485 – 488)

Review comment:  I would suggest that it should mainly take up:

The salient review’s findings and their implications for the policy makers/ planners and researchers.

Response: We believe these are covered throughout the discussion.

The limitations.

Response: These have been covered in the “Study limitations and strengths” section.

The impact of the changes in the registered protocol, if any, should be mentioned.

Response: We have added the following text: “we made some changes to our original protocol as reported in S2 Appendix. These changes allowed us to be more inclusive of potentially relevant literature given the paucity of studies conducted in slum settings, while being unlikely to introduce particular bias to the review.” 

The statement may be found on Pages 39 and 40; Lines 690 – 692.

Age group distribution; and decayed, missing and filled components of dental caries-implications???.

Response: We have added the following text: “Dental caries affected all ages but were noted to increase with higher WHO index age categories. The values for the “Missing” component of the DMFT were usually the highest, followed by the “Decayed” component. The values for the “Filled” component were the lowest in all measures. These reflected the lack of oral health surveillance and access to dental services, leading to ongoing decay and loss of teeth before treatment could be obtained, highlighting the need for better access to dental healthcare in LMICs and particularly in slum setting In future population-based oral health surveys, a second caries-measuring tool – the Significant Caries (SiC) index – might be used together with DMFT to reflect the situation of the most caries-exposed individuals [47]. Information thus provided would enable countries to direct resources to the groups that are worst affected [47, 48]. 

The statement may be found on Pages 34 – 35; Lines 556 – 565.

Gingivitis/ bleeding gums (reversible and self-manageable conditions). The need for oral health education should be recommended).

Response: We have added the following text: Among the sub-categories of periodontal disease, bleeding gum/ gingivitis formed the highest prevalence in most countries. The fact that these conditions are reversible and self-manageable has implications for targeted oral health education among affected population groups [50].

The statement may be found on Page 35; Lines 567 – 570

The use of fluoride, explored in very few studies, should be stressed upon.

Response: We have added the following text: Access to fluoride is crucial in preventing dental diseases [51, 52], yet there is a sparse of studies on populations' exposure to fluoride from sources like toothpaste, food, water, and applications in the LMICs. This information is necessary for understanding oral health risks and planning public health interventions effectively.

 The statement may be found on Page 35; Lines 631 – 635

Review comment:  Structural barriers to the availability, accessibility and use of dental services (publicly funded/ private, insurance etc.) should be brought to planners' attention.

Response: Thank you. We have highlighted this in the discussion. (Page 38; Lines 652 – 655)

Review comment:  The last three are related to lines 114-116 which need to be rephrased as well.

Response: We are unclear about what the reviewer is referring to here due to the mismatch in quoted line numbers compared with our manuscript.

Review comment:  In addition, in Line 192, exposure/ outcome (the terms more related to the experimental studies) are confusing???

Response: (From Line 167, page 7) Our understanding is that PICO framework is more related to experimental studies where the ‘I’ stands for intervention. PECO framework is a well-recognized framework that is suitable for the formulation of research questions that explore the environmental impact on health and related outcomes (for example, see Morgan et al. Identifying the PECO: A framework for formulating good questions to explore the association of environmental and other exposures with health outcomes. Environ Int 2018;121(Ot 1):1027-1031, https://www.ncbi.nlm.nih.gov/pmc/articles/PMC6908441/ ). As our review explores potential impacts of living in slums on dental disease and oral health practices, we believe it is a suitable framework to use.

Review comment:  Lines 193-194/ 230-232: Who trained the reviewers and how???

Response: We have now explained this, thank you. (Page 10; Line 222)

Review comment:  Words to replace: Lines 246-accessed or assessed? Lines 402-403 regular dental visits or irregular ones???

Response: Thank you. We have changed ‘accessed’ to ‘assessed’ (Page 12; Line 282). The statement concerning regular visits to the dentist (Page 25; Lines 437-439) was based on findings from Costa et al. (presented in Table 3), in which a greater prevalence of high caries severity was found among those who frequently visited the dentist (prevalence ratio [PR] = 1.18; 95%CI: 1.07 to 1.30) in comparison to those who did not make regular visits to the dentist. We believe this could be explained by reverse causation (higher severity of dental caries leading to more visits to the dentist) and therefore have changed our wording from ‘risk factors for dental diseases’’ to ‘factors associated with dental diseases’ (Lines 437-439) to avoid giving out the impression that the listed factors necessarily ‘cause’ dental diseases.

---

## [Decision Letter · Decision Letter 2]

9 Aug 2024

Systematic review of oral health in slums and non-slum urban settings of Low and Middle-Income Countries (LMICs): disease prevalence, determinants, perception, and practices

PONE-D-23-35856R2

Dear Dr. Mary Ebelechukwu Osuh, 

We’re pleased to inform you that your manuscript has been judged scientifically suitable for publication and will be formally accepted for publication once it meets all outstanding technical requirements.

Kind regards,

Tanay Chaubal

Academic Editor

PLOS ONE

Additional Editor Comments (optional):

Reviewers' comments:

Reviewer's Responses to Questions

**Comments to the Author**

1. If the authors have adequately addressed your comments raised in a previous round of review and you feel that this manuscript is now acceptable for publication, you may indicate that here to bypass the “Comments to the Author” section, enter your conflict of interest statement in the “Confidential to Editor” section, and submit your "Accept" recommendation.

Reviewer #3: All comments have been addressed

2. Is the manuscript technically sound, and do the data support the conclusions?

Reviewer #3: Yes

3. Has the statistical analysis been performed appropriately and rigorously? 

Reviewer #3: Yes

4. Have the authors made all data underlying the findings in their manuscript fully available?

Reviewer #3: Yes

5. Is the manuscript presented in an intelligible fashion and written in standard English?

Reviewer #3: Yes

6. Review Comments to the Author

Reviewer #3: I am sorry for the mismatch regarding Line numbers but I followed the numbers which were there in the version of the manucript I downloaded from the Journal's link sent to me by the editorial office.

7. PLOS authors have the option to publish the peer review history of their article (what does this mean?). If published, this will include your full peer review and any attached files.

Reviewer #3: **Yes: **Haleem A

---

## [Editor Report · Acceptance letter]

2 Sep 2024

PONE-D-23-35856R2 

PLOS ONE

Dear Dr. Osuh, 

I'm pleased to inform you that your manuscript has been deemed suitable for publication in PLOS ONE. Congratulations! Your manuscript is now being handed over to our production team.

Kind regards, 

on behalf of

Dr. Tanay Chaubal 

Academic Editor

PLOS ONE